# Generalized Gibbs Ensemble and string-charge relations in nested Bethe Ansatz

György Z. Fehér[1] and Balázs Pozsgay[2]

**1** BME Statistical Field Theory Research Group 1111 Budapest, Budafoki út 8, Hungary
**2** MTA-BME Quantum Dynamics and Correlations Research Group,
Department of Theoretical Physics, Budapest University of Technology and Economics,
1521 Budapest, Hungary

## Abstract

The non-equilibrium steady states of integrable models are believed to be described by the Generalized Gibbs Ensemble (GGE), which involves all local and quasi-local conserved charges of the model. In this work we investigate integrable lattice models solvable by the nested Bethe Ansatz, with group symmetry $SU(N)$, $N \geq 3$. In these models the Bethe Ansatz involves various types of Bethe rapidities corresponding to the "nesting" procedure, describing the internal degrees of freedom for the excitations. We show that a complete set of charges for the GGE can be obtained from the known fusion hierarchy of transfer matrices. The resulting charges are quasi-local in a certain regime in rapidity space, and they completely fix the rapidity distributions of each string type from each nesting level.


# 1  Introduction

One of the central problems in theoretical physics is the connection between the fundamental laws of quantum mechanics and the various classical theories describing physics on macroscopic scales. A particularly interesting question is equilibration and thermalization of closed quantum systems, i.e. the emergence of statistical mechanics from the unitary time evolution dictated by the Schrödinger equation. This problem has attracted interest since the 1930's, and significant understanding has been achieved in the last 15 years (for reviews, see [1,2]). Furthermore, special attention has been devoted to those systems which do not thermalize, and one class of such systems are the integrable models.

One dimensional exactly solvable models are known to possess an infinite number of mutually commuting conserved charges. The resulting conservation laws prevent the integrable systems from thermalization to standard Gibbs ensembles. Instead, the idea of the Generalized Gibbs Ensemble (GGE) was put forward in [3,4]. The GGE is analogous to the canonical Gibbs ensemble: it is built on the maximum entropy principle [5], but it involves all the conserved charges of the model.

Even though the concept of the GGE is only $\sim 10$ years old, it has a quite rich history. Whereas it was fairly quickly proven to be the correct thermodynamic ensemble in the case of free systems [6–19], the case of interacting lattice models had its twists and turns. After some early to attempts to construct the GGE for the Heisenberg spin chain [20, 21] it was shown in [22,23] that the GGE built on the known set of strictly local charges fails to give correct predictions for the steady states in particular quench situations. This failure was later attributed to an *incompleteness* of the known charges, and the work [24] showed that a *Complete GGE* can be built by incorporating the recently discovered quasi-local charges [25, 26] as well.

After clarifying the GGE for the Heisenberg chains it became widely accepted that there should be a complete GGE for any integrable model, and the remaining issue is to find the correct set of conserved charges. Whereas this might seem like a relatively minor problem, it is far from being trivial in models more complicated than the XXZ spin chain. Ultimately one would like a general proof for the existence of a Complete GGE, at least after specifying the integrability structure of the model. However, such a proof is not yet in sight.

The theory of Generalized Hydrodynamics (GHD) also motivates the further study of the GGE. GHD describes large scale transport properties of integrable models [27, 28], and one

of the main assumptions of the theory is the existence of local (space and time dependent) GGE's, for which there exists a complete set of charges. The GHD has been applied already for more complicated systems such as the Hubbard model [29], where this completeness has not yet been studied. Therefore it is important to understand the GGE in these more complicated models.

Here we contribute to the subject by formulating the GGE for a prototypical multi-component system, namely the $SU(3)$-symmetric fundamental spin chain, also known as the Lai-Sutherland model. Furthermore, we present some conjectures about the GGE in the fundamental $SU(N)$-symmetric model for any $N \geq 3$. These models can be solved by the so-called nested Bethe Ansatz [30–33]. The eigenstates can be characterized by multiple sets of rapidities: the first set describes the lattice momenta of the quasi-particles, which are the excitations above a reference state, whereas the remaining sets describes the wave function amplitudes in the internal space of the spin waves.

We should note that even though the GGE for these particular models has not yet been set up, specific quantum quenches in the $SU(3)$ case have already been studied. They all involve so-called integrable initial states [34], which allow an exact analytic solution due to certain relations to boundary integrability [35]. A quantum quench from a specific Matrix Product State (MPS) was already studied in [36], and local two-site states were investigated in [37, 38]. The light cone spreading of entanglement and correlations was studied in [39]. Nevertheless the question of the existence of a complete GGE in the $SU(3)$-symmetric model remained completely open up to now.

In the following subsection we give a more detailed description of the GGE in generic integrable models and explain the main mechanisms responsible for the emergence of the GGE, while omitting many technical details. Afterwards, the remainder of the paper is organized as follows: In section 2 we define the $SU(N)$ symmetric spin chain, and consider the most basic properties of it. In section 3 we discuss the GGE in the $N = 2$ case, which is the celebrated XXX Heisenberg spin chain. This Section includes known results, but we re-derive them using slightly different techniques, more adequate for later generalizations. Section 3 thus sets the stage for our investigations of the multi-component models. In section 4 we introduce the main interest of our paper, the $SU(3)$ symmetric model, and discuss its main properties. In section 5 we consider two generating functions for conserved charges, which correspond to the defining and conjugate representations of $SU(3)$. We rigorously prove their quasi-locality. In section 6 we build two families of charges, and we derive the complete set of string-charge relations for this model. In section 7 we conjecture the structure of the complete GGE for generic $SU(N)$. We conclude in 8, and a number of technical computations are collected in the appendices.

## 1.1 Generalized Eigenstate Thermalization

Let us consider an integrable lattice model with a local Hamiltonian $H$, defined in some finite volume $L$.

We consider a quantum quench situation, when the system is prepared in the initial state $|\Psi(t=0)\rangle = |\Psi_0\rangle$. We will consider different volumes and eventually the thermodynamic limit. Therefore we require, that $|\Psi_0\rangle$ should be well-defined in any finite volume and also as $L \to \infty$. One possibility is to define $|\Psi_0\rangle$ as the ground state of a different local Hamiltonian, or as a Matrix Product State (MPS) [40]. We also require that $|\Psi_0\rangle$ satisfies the cluster decomposition principle, namely for two local operators $\mathcal{O}_{1,2}(x)$

$$\lim_{|x_1 - x_2| \to \infty} \langle \Psi_0 | \mathcal{O}_1(x_1) \mathcal{O}_2(x_2) | \Psi_0 \rangle = \langle \Psi_0 | \mathcal{O}_1(x_1) | \Psi_0 \rangle \langle \Psi_0 | \mathcal{O}_2(x_2) | \Psi_0 \rangle. \tag{1}$$

The state of the system at later times is given by

$$|\Psi(t)\rangle = e^{-iHt}|\Psi_0\rangle. \tag{2}$$

We are interested in equilibration and thermalization, to this order we investigate the long time limit of local observables. Here and in the following we understand that the $L \to \infty$ is taken *before* the long time limit. However, certain formal expressions are more easily handled by keeping $L$ finite.

Before turning to the integrable models, let us focus on the more simple non-integrable case. In generic non-integrable systems equilibration and thermalization to a Gibbs Ensemble can be argued as follows [1].

A direct finite volume expansion for the time evolution gives

$$\langle \mathcal{O}(t) \rangle = \sum_{j,k} \langle \Psi_0|\Psi_j\rangle \langle \Psi_j|\mathcal{O}|\Psi_k\rangle \langle \Psi_k|\Psi_0\rangle e^{-i(E_k-E_j)t}. \tag{3}$$

In the long time limit dephasing leads to the emergence of the Diagonal Ensemble:

$$\lim_{T\to\infty}\left[\int_0^T dt\, \langle \mathcal{O}(t)\rangle\right] = \sum_j |\langle \Psi_0|\Psi_j\rangle|^2 \langle \Psi_j|\mathcal{O}|\Psi_j\rangle. \tag{4}$$

The Eigenstate Thermalization Hypothesis (ETH) states that for almost all states in a small energy window $[E, E + \Delta E]$ the mean values $\langle \Psi_j|\mathcal{O}|\Psi_j\rangle$ will be close to each other [41, 42]. Due to energy conservation the system will be populated only with states that are close to each other in energy density, therefore the diagonal ensemble has to be equal to the microcanonical average. In large volumes the microcanonical and canonical averages become equivalent for local operators, thus we have argued for the emergence of the Gibbs Ensemble:

$$\lim_{T\to\infty}\left[\int_0^T dt\, \langle \mathcal{O}(t)\rangle\right] = \langle \mathcal{O}\rangle_{\text{GE}} \equiv \frac{\text{Tr}\left(e^{-\beta H}\mathcal{O}\right)}{\text{Tr}\left(e^{-\beta H}\right)}. \tag{5}$$

Here the parameter $\beta$ has to be chosen such that energy conservation holds:

$$\langle \Psi_0|H|\Psi_0\rangle = \langle H\rangle_{\text{GE}}. \tag{6}$$

In integrable models the situation is different due to the existence of a large family of additional conserved charges. It has been known since the early days of integrability that the integrable lattice model possesses a family of commuting operators

$$[\mathcal{Q}_j, \mathcal{Q}_k] = 0, \qquad j, k = 1, \dots, \infty, \tag{7}$$

such that the Hamiltonian is a member of the series and each $\mathcal{Q}_k$ is an extensive operator whose operator density is strictly local. Typically it is possible to choose the charges such that the density of $\mathcal{Q}_k$ spans $k$ sites.

The existence of these charges leads to the concept of the Generalized Gibbs Ensemble (GGE). The main idea is to involve all conservation laws in the standard statistical physical derivations. Based on the maximum entropy principle we expect that the equilibrated values of local observables will be given by

$$\langle \mathcal{O}\rangle_{\text{GGE}} \equiv \frac{\text{Tr}\left(e^{-\sum_j \beta_j \mathcal{Q}_j}\mathcal{O}\right)}{\text{Tr}\left(e^{-\sum_j \beta_j \mathcal{Q}_j}\right)}. \tag{8}$$

Here the generalized inverse temperatures $\beta_j$ are determined by the initial state through

$$\langle \Psi_0 | \mathcal{Q}_j | \Psi_0 \rangle = \big\langle \mathcal{Q}_j \big\rangle_{\text{GGE}}, \qquad j = 1, \ldots, \infty, \tag{9}$$

which are a set of coupled non-linear equations.

In analogy with the non-integrable case, where the ETH is the main mechanism for the emergence of the GE, in integrable models equilibration to the GGE is guaranteed if the Generalized Eigenstate Thermalization (GETH) holds with the given set of conserved charges [43]. In rough terms the GETH states that in the TDL the mean values of local observables only depend on the global mean values of the conserved charges, and not on any other details of the state.

It was realized in the case of the Heisenberg spin chains that the GETH does not hold if we only consider the traditional set of local charges [22, 23, 44, 45]. Instead, it was realized that the so-called quasi-local charges [25, 26] need to be included as well [24, 46]. The main reason for this is the following.

In integrable models solvable by the Bethe Ansatz the finite volume eigenstates are characterized by a finite set of Bethe rapidities. In the thermodynamic limit the equilibrium configurations are described by root distribution functions $\rho_\alpha(\lambda)$, where $\lambda$ is the rapidity parameter and $\alpha$ is an index or multi-index describing particle types. It is a general understanding that in such models the local correlation functions depend on all root densities; this was proven for the XXZ chain in [47, 48]. According to this picture, a set of conserved operators is complete, *if their mean values completely fix all the Bethe root densities*. This is the ultimate form of the GETH, relevant for interacting integrable models. This idea was further formalized in [49, 50], where it was argued that the GGE should be formulated using root density operators, whose eigenvalues are the root densities themselves.

In Section 3 we summarize the known results of the XXX chain and show that a complete set of quasi-local charges indeed fixes all the root densities. It is the goal of our paper to extend this picture to the $SU(N)$-symmetric chains with $N \geq 3$, and to find a complete set of quasi-local operators.

We will show that similar steps are needed also in the $SU(3)$-symmetric model. That model has a more complicated Bethe Ansatz solution and corresponding fusion hierarchy of transfer matrices, nevertheless the inversion relations take an identical form, and are equally important for the derivations of the string-charge relations.

## 2 The $SU(N)$-symmetric spin chains - generalities

Let us consider an integer $N \geq 2$. We define a spin chain with local Hilbert spaces $h_j = \mathbb{C}^N$ called quantum spaces such that the full Hilbert space of the chain if length $L$ is $\mathcal{H}_L = h_1 \otimes h_2 \otimes \cdots \otimes h_L = \left( \mathbb{C}^N \right)^L$.

We consider the fundamental $SU(N)$-symmetric model [54, 55] defined on this Hilbert space, given by the Hamiltonian

$$H = -L + \sum_{j=1}^{L} P_{j,j+1}. \tag{10}$$

Above $P \in \text{End}(\mathbb{C}^N \otimes \mathbb{C}^N)$ is the permutation operator, which acts as $P(v_1 \otimes v_2) = v_2 \otimes v_1$, $v_1, v_2 \in \mathbb{C}^N$. For simplicity we consider the model under periodic boundary conditions: $P_{L,L+1} = P_{L,1}$.

For $N = 2$ the model is equivalent to the famous Heisenberg XXX spin chain, whereas for $N \geq 3$ it can be considered a higher rank generalization of it.

One of the most important properties of the Hamiltonian (10) is its $SU(N)$ invariance, which is understood as follows. Let the local Hilbert spaces $h_j$ carry the defining representation of $SU(N)$, and let us extend the group action to the tensor product. Then the global $SU(N)$ invariance of the Hamiltonian immediately follows from the fact that it involves invariant local operators.

The model is integrable for any $N$: it possesses an infinite family of commuting local charges, and it can be solved by the Algebraic Bethe Ansatz. The exact real space wave functions of the eigenstates are given by the so-called nested Bethe Ansatz [30–33]. In the following we briefly review the standard integrability framework of these models. We focus on the construction of the commuting set of transfer matrices, and their eigenvalues expressed in terms of Bethe Ansatz rapidities. We do not treat the actual construction of the Bethe states, and we refer the reader to [56, 57].

Let us consider the following fundamental $R$-matrix:

$$R(u) = \frac{1}{u+i}(u+iP), \qquad R \in \text{End}(\mathbb{C}^N \otimes \mathbb{C}^N), \tag{11}$$

which satisfies the Yang-Baxter equation [58]

$$R_{23}(v-w)R_{13}(u-w)R_{12}(u-v) = R_{12}(u-v)R_{13}(u-w)R_{23}(v-w), \tag{12}$$

and the unitarity relation:

$$R(u)R(-u) = 1. \tag{13}$$

It is group invariant with respect to $GL(N)$:

$$G_1 G_2 R(u) = R(u) G_1 G_2, \quad G_i \in GL(N), i = 1, 2. \tag{14}$$

Let us consider an additional space $h_0 = \mathbb{C}^N$ called the auxiliary space. We define the transfer matrix (TM) of the model in the usual way:

$$t(u) = \text{Tr}_0 R_{10}(u) R_{20}(u) \dots R_{L0}(u), \tag{15}$$

where the trace is taken on the auxiliary space $h_0$. The transfer matrices form a commuting family:

$$[t(u), t(v)] = 0. \tag{16}$$

The commuting set of local charges is built from $t(u)$. Let

$$\mathcal{Q}_k = (-i) \left. \frac{d^{k-1}}{du^{k-1}} \log t(u) \right|_{u=0}, \qquad k \geq 2. \tag{17}$$

It can be shown that the $\mathcal{Q}_k$ are local charges: they are extensive such that their operator density spans at most $k$ sites [59]. It follows from (16) that they commute with each other:

$$[\mathcal{Q}_j, \mathcal{Q}_k] = 0, \qquad j, k \geq 2. \tag{18}$$

Furthermore, the Hamiltonian is a member of this series. Direct computation gives $H = -\mathcal{Q}_2$.

In our work a special role will be played by the so-called fusion hierarchy of the transfer matrices. In the following we briefly introduce these concepts, while omitting many technical details.

Let $\Lambda_1$ and $\Lambda_2$ be two irreducible representations of $SU(N)$. It is known that there exists an $R$-matrix $R^{\Lambda_1,\Lambda_2}(u)$ acting on the tensor product of the two representations, which is unique up to an overall scaling and certain shifts in the rapidity parameter, such that for any three representations $\Lambda_j$, $j = 1, 2, 3$ they satisfy the Yang-Baxter equation [60]:

$$R_{23}^{\Lambda_2,\Lambda_3}(v-z)R_{13}^{\Lambda_1,\Lambda_3}(u-z)R_{12}^{\Lambda_1,\Lambda_2}(u-v) = R_{12}^{\Lambda_1,\Lambda_2}(u-v)R_{13}^{\Lambda_1,\Lambda_3}(u-z)R_{23}^{\Lambda_2,\Lambda_3}(v-z). \tag{19}$$

These $R$-matrices can be obtained by the so-called fusion procedure [60, 61].

In our models each local spin variable carries the defining representation of $SU(N)$, therefore we will need $R$-matrices acting on the tensor product of the defining representation and some other $\Lambda$. For these cases we will use the short notation $R_{12}^{\Lambda}(u)$.

For each representation $\Lambda$ we define the transfer matrix with auxiliary space carrying $\Lambda$ as

$$t^{\Lambda}(u) = \mathrm{Tr}_0 R_{10}^{\Lambda}(u) R_{20}^{\Lambda}(u) \dots R_{L0}^{\Lambda}(u). \tag{20}$$

It follows from (19) that all of these transfer matrices commute:

$$[t^{\Lambda}(u), t^{\Lambda'}(v)] = 0. \tag{21}$$

The representations of $SU(N)$ can be described by Young diagrams. The transfer matrices corresponding to rectangular Young diagrams play a special role in the theory, and we will show that they are central also for the GETH. For the Young diagram with $a$ rows and $s$ columns the corresponding transfer matrix will be denoted as $t_s^{(a)}(u)$. These objects satisfy a closed set of functional relations called the Hirota equation or $T$-system relations; specific details will be given later, and for reviews see [62, 63].

The common eigenstates of the transfer matrices can be found by the (nested) Bethe Ansatz. The actual construction, and hence the discussion of the GGE and the GETH strongly depends on $N$. In the next section we review the known results for the case $N = 2$, which corresponds to the XXX Heisenberg spin chain. In 4 we start our discussion of the $SU(3)$-symmetric chain, which is the main subject of this paper. However, before going to these special cases we introduce the notion of quasi-local charges, that are essential for the GETH.

## 2.1 Quasi-local charges

The set of canonical local charges (17) has been known since the early days of integrability. On the other hand, the existence and importance of quasi-local charges was only understood in recent years [25, 26]. Here we define quasi-locality following the works [25, 26].

Consider the physical Hilbert space $\mathcal{H}_L = h_1 \otimes \dots \otimes h_L = \left(\mathbb{C}^N\right)^L$, and consider $\mathrm{End}(\mathcal{H}_L)$, the space of linear operators over $\mathcal{H}_L$. $\mathrm{End}(H_L)$ possesses a Hilbert space structure, under the Hilbert-Schmidt scalar product, defined as

$$\langle A, B \rangle_{\mathrm{HS}} = N^{-L} \mathrm{Tr}\left(A^{\dagger} B\right), \quad A, B \in \mathrm{End}\left(\mathcal{H}_L\right). \tag{22}$$

The Hilbert-Schmidt norm is defined as

$$\|A\|_{\mathrm{HS}}^2 = \langle A, A \rangle_{\mathrm{HS}} = N^{-L} \mathrm{Tr}\left(A^{\dagger} A\right). \tag{23}$$

This normalization is such, that for the identity operator $\|1\|_{\mathrm{HS}} = 1$.

We define the traceless part of an operator $A$ as

$$\{A\} = A - N^{-L} \mathrm{Tr}(A). \tag{24}$$

Quasi-locality is defined for traceless operators.

The $L$-dependent operator $\{A\}(L) \in \mathcal{H}_L$ is called quasi-local, if it satisfies the following properties:

1. $\{A\}(L)$ is translationally invariant for every $L$.

2. In large volumes $\|\{A\}\|_{\mathrm{HS}}^2 \sim L$

3. For any locally supported $k$-site operator $b = b_k \otimes 1_{L-k}$ the overlap $\langle b, \{A\} \rangle_{\mathrm{HS}}$ is asymptotically independent of $L$ in the $L \to \infty$.

Note that the quasi-locality only makes sense in the strictly $L \to \infty$ limit, because it is the property of the infinite series of operators $\{A\}(L)$. The strictly local charges obviously satisfy these requirements.

A quasi-local operator can be written in the form

$$A(L) = \sum_{x=1}^{L} a(x), \tag{25}$$

where $a(x)$ is the operator density of $A$. It does not have to be local, but it has to have a finite norm. As an effect, the long-range contributions to $a(x)$ have (typically exponentially) decreasing amplitudes.

## 3 The Heisenberg spin chain

The $SU(2)$-symmetric Heisenberg XXX spin chain is defined conventionally by the Hamiltonian

$$H_{XXX} = \sum_{j=1}^{L} \left( \sigma_j^x \sigma_{j+1}^x + \sigma_j^y \sigma_{j+1}^y + \sigma_j^z \sigma_{j+1}^z - 1, \right), \tag{26}$$

where $\sigma_j^\alpha$ are the Pauli matrices acting on quantum space $j$. In this normalization $H_{XXX} = 2H$, where $H$ is the Hamiltonian (10) at $N = 2$.

The exact eigenstates of this model were first found by Bethe [64]. They can be characterized by a set of rapidities $\{\lambda_1, \ldots, \lambda_N\}$, which parametrize the quasi-momenta of interacting spin waves above the reference state $|\emptyset\rangle$, which is chosen conventionally as the state with all spins up. The un-normalized eigenstates can be written as

$$|\boldsymbol{\lambda}_N\rangle = \sum_{1 \leq n_1 < \ldots < n_N \leq L} \Psi(n_1, \ldots, n_N) \prod_{s=1}^{N} (\sigma_-)_{n_s} |\emptyset\rangle,$$

$$\Psi(n_1, \ldots, n_N) = \sum_{\mathcal{P} \in \mathcal{S}^N} \left( \prod_{1 \leq r < l \leq N} \frac{\lambda_{\mathcal{P}(l)} - \lambda_{\mathcal{P}(r)} - \mathrm{i}}{\lambda_{\mathcal{P}(l)} - \lambda_{\mathcal{P}(r)}} \prod_{r=1}^{N} \left( \frac{\lambda_{\mathcal{P}(r)} + \frac{\mathrm{i}}{2}}{\lambda_{\mathcal{P}(r)} - \frac{\mathrm{i}}{2}} \right)^{n_r} \right). \tag{27}$$

Here $\sigma_-$ is the spin lowering operator, and the $n_s$ describe the positions of the spin waves.

These states are eigenstates of the Hamiltonian if the wave functions are periodic, which leads to the Bethe equations:

$$\left( \frac{\lambda_j + \frac{\mathrm{i}}{2}}{\lambda_j - \frac{\mathrm{i}}{2}} \right)^L = \prod_{k=1, k \neq j}^{N} \frac{\lambda_j - \lambda_k + \mathrm{i}}{\lambda_j - \lambda_k - \mathrm{i}} \qquad j = 1, \ldots, N. \tag{28}$$

The lattice momentum and the energy of a Bethe state $\left| \{\lambda_j\}_{j=1}^N \right\rangle$ are given by the sum of one particle momentum and energy, respectively:

$$P = \sum_{j=1}^{N} p(\lambda_j), \qquad\qquad p(\lambda) = \mathrm{i} \log \left( \frac{\lambda_j + \frac{\mathrm{i}}{2}}{\lambda_j - \frac{\mathrm{i}}{2}} \right), \tag{29}$$

$$E = \sum_{j=1}^{N} \varepsilon(\lambda_j), \qquad\qquad \varepsilon(\lambda) = -\frac{1}{\lambda^2 + \frac{1}{4}}. \tag{30}$$

The eigenvalues of the fundamental transfer matrix (15) can be computed using Algebraic Bethe Ansatz [65]. On the Bethe state $|\{\lambda\}_N\rangle$ the eigenvalues are

$$t(u) = \frac{Q_1\left(u - \frac{i}{2}\right)}{Q_1\left(u + \frac{i}{2}\right)} + \frac{Q_0(u)Q_1\left(u + \frac{3i}{2}\right)}{Q_0(u + i)Q_1\left(u + \frac{i}{2}\right)}, \tag{31}$$

where

$$Q_0(u) = u^L, \qquad Q_1(u) = \prod_{j=1}^{N}(u - \lambda_j). \tag{32}$$

The Bethe states are highest weight with respect to the global $SU(2)$ symmetry; other states in the same multiplet can be obtained by global spin lowering operators.

## 3.1 String hypothesis and Thermodynamic Bethe Ansatz

In order to study the thermodynamic limit of the model it is important to know the positions of the Bethe roots in the complex plain. It is known that in Bethe Ansatz solvable models the roots typically arrange themselves into so-called string patterns. A string describes a bound state of spin waves. The string hypothesis states that in the thermodynamic limit the dynamical processes can be described by concentrating only on the regular string solutions, neglecting contributions from the rare outlier states.

In the XXX model an $n$-string pattern centered around the rapidity $x \in \mathbb{R}$ takes the form

$$x^\ell = x + i\left(\frac{n+1}{2} - \ell\right) + \delta^\ell, \quad \ell = 1, \dots, n. \tag{33}$$

Here $\delta^\ell$ are the so-called string deviations which are exponentially small in large volumes for regular string solutions.

A Bethe state with $N_n$ number of $n$-strings thus consists of the rapidities

$$\lambda_\alpha^{n,\ell} = \lambda_\alpha^n + i\left(\frac{n+1}{2} - \ell\right) + \delta_\alpha^{n,\ell}, \quad \ell = 1, \dots, n, \quad \alpha = 1, \dots, N_n. \tag{34}$$

The total number of Bethe roots is computed simply as

$$\sum_{n=1}^{\infty} n N_n = N. \tag{35}$$

We will be interested in the thermodynamic limit, when $N \to \infty$, $L \to \infty$ with a fixed $N/L$ ratio. In this limit the string deviations become exponentially small and according to the string hypothesis it is sufficient to describe the positions of the string centers. Then the string centers can be described by continuous density functions along the real line. These are denoted by $\rho_n(u)$, and are normalized such that in a large volume $L$ the total number of $n$-strings between rapidities $u$ and $u + \Delta u$ is $L\rho_n(u)\Delta u$.

Analogously to a free system we also introduce hole densities. For a Bethe state a hole is a position in rapidity space which would satisfy the Bethe equations but it is not actually a Bethe root. Such holes can be defined for each particle type and each string pattern, and in the TDL they are described by the densities $\rho_{h,n}$.

In the thermodynamic limit the Bethe equations (73) can be transformed into a set of coupled linear integral equations:

$$\rho_{t,n}(u) = \delta_{n,1}s(u) + s \star \left(\rho_{h,n-1}^{(1)} + \rho_{h,n+1}^{(1)}\right)(u), \tag{36}$$

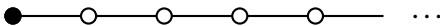

Figure 1: TBA diagram of the XXX model. Each node corresponds to a string type, and the links denote the convolution kernels in Eq. (36). The filling of the first node signals the source term for $\rho_{t,1}(u)$.

where $\rho_{t,n}^{(j)}$, $j = 1, 2$ are the so-called total root densities, defined by

$$\rho_{t,n}^{(j)}(u) = \rho_n^{(j)}(u) + \rho_{h,n}^{(j)}(u), \qquad j = 1, 2. \tag{37}$$

Furthermore, the convolution is understood as

$$(f \star g)(\lambda) = \int_{-\infty}^{\infty} d\mu \, f(\lambda - \mu) g(\mu), \tag{38}$$

and the integration kernel is

$$s(\lambda) = \frac{1}{2 \cosh(\pi \lambda)}. \tag{39}$$

The integral equations (36) are symbolically depicted on Fig. 1.

## 3.2 String-charge relations

The main question of the GETH is to what extent a given set of charges determines the Bethe root densities. Let us first focus on the set of strictly local charges $\{\mathcal{Q}_a\}$ defined in (17). Instead of dealing with the discrete set it is useful to consider the generating function $X_1(u)$ defined formally as

$$X_1(u) = (-\mathrm{i})\partial_u \log t(u) = \sum_{j=2}^{\infty} \frac{u^{j-2}}{(j-2)!} \mathcal{Q}_j. \tag{40}$$

It was shown in [21, 22] that the eigenvalues of this operator are asymptotically

$$\frac{1}{2\pi L} X_1(u) = s \star (\rho_{h,1} + a_1), \tag{41}$$

where

$$a_1(u) = \frac{1}{2\pi} \frac{1}{u^2 + \frac{1}{4}}. \tag{42}$$

It follows that this set of charges is not sufficient to determine all Bethe root densities: $X_1(u)$ only fixes the hole density of the 1-strings.

This situation was remedied in [24] (see also [46, 49]), where it was shown that the recently introduced quasi-local charges [25, 26] contain just enough information to fix all the root densities.

The quasi-local charges are obtained from the fusion hierarchy of the transfer matrices. Let us define the higher spin Lax operators with spin $s = m/2$, $m \in \mathbb{N}$ acting on the tensor product $V_a \otimes V_j = \mathbb{C}^{2s+1} \otimes \mathbb{C}^2$ as

$$\mathcal{L}_{a,j}^m(u) = \frac{u + \mathrm{i}\frac{1}{2} + \mathrm{i}\boldsymbol{S}_a \cdot \boldsymbol{\sigma}_j}{u + \mathrm{i}\frac{m+1}{2}}, \tag{43}$$

where $\boldsymbol{S}_a$ stands for the vector of the spin-$s$ generators of $SU(2)$, and $\boldsymbol{\sigma}_j$ is the vector constructed out of Pauli matrices. Our conventions for the Lax operators differs slightly from the one used in [24, 25, 46].

We define the corresponding transfer matrices (TMs)

$$t_m(u) = \mathrm{Tr}_a \mathcal{L}^m_{a,1}(u)\mathcal{L}^m_{a,2}(u)\dots\mathcal{L}^m_{a,L}(u). \tag{44}$$

It can be shown that these operators form a commuting family:

$$[t_m(u), t_n(v)] = 0. \tag{45}$$

The spin-$s$ representations correspond to Young diagrams with 1 row and $m$ columns, therefore using the notations of Section 2 we have the identification $t_m(u) = t_m^{(1)}(u)$.

These transfer matrices satisfy a set of functional equations called the Hirota equation or $T$-system:

$$t_m\left(u+\frac{\mathrm{i}}{2}\right)t_m\left(u-\frac{\mathrm{i}}{2}\right) = t_{m+1}(u)t_{m-1}(u) + \phi_m(u). \tag{46}$$

Here $\phi_m(u)$ is a scalar function (independent of the Bethe state) given by

$$\phi_m(u) = \frac{Q_0\left(u-\mathrm{i}\frac{m}{2}\right)}{Q_0\left(u+\mathrm{i}\frac{im}{2}\right)}. \tag{47}$$

Furthermore, the initial value for the recursion is $t_0(u) = 1$.

The eigenvalues on the common eigenstates are [66]

$$t_m(u) = \frac{Q_1\left(u-\mathrm{i}\frac{m}{2}\right)Q_1\left(u+\mathrm{i}\frac{m+2}{2}\right)}{Q_0\left(u+\mathrm{i}\frac{m+1}{2}\right)}\sum_{k=0}^{m}\frac{Q_0\left(u+\mathrm{i}\frac{m+1}{2}-\mathrm{i}k\right)}{Q_1\left(u+\mathrm{i}\frac{m}{2}-\mathrm{i}k\right)Q_1\left(u+\mathrm{i}\frac{m+2}{2}-\mathrm{i}k\right)}. \tag{48}$$

It can be checked by direct computation that these eigenvalues satisfy the $T$-system (46).

A key role is played by the operators $X_m(u)$ defined formally by

$$X_m(u) = (-\mathrm{i})\partial_u \log t_m(u). \tag{49}$$

It was shown in [25] that the traceless operators $\{X_m(u)\}$ are quasi-local in the thermodynamic limit, if $u$ is within the physical strip $\mathcal{P}$ defined as

$$\mathcal{P} \equiv \left\{u \in \mathbb{C}, \quad |\Im(u)| < \frac{1}{2}\right\}. \tag{50}$$

For a more precise treatment of $X_m(u)$ see the next subsection.

Regarding the eigenvalues of the operators $X_m(u)$ it was obtained in [24]

$$\rho_{h,n} = a_n - \frac{1}{2\pi L}(X_n^{[+]} + X_n^{[-]}), \tag{51}$$

where

$$a_n(\lambda) = \frac{1}{2\pi}\frac{n}{\lambda^2 + \frac{n^2}{4}}, \tag{52}$$

and we introduced the short-hand notation:

$$f^{[\pm]}(u) = \lim_{\varepsilon \to 0} f\left(u \pm \frac{\mathrm{i}}{2} \mp \varepsilon\right). \tag{53}$$

Making use of the system (36) an equivalent form can be derived:

$$\rho_n = \frac{1}{2\pi L}\left(X_n^{[+]} + X_n^{[-]} - X_{n-1} - X_{n+1}\right). \tag{54}$$

Thus the higher spin transfer matrices contain just enough information to determine all the root densities.

Let us comment on some important differences between the finite volume situation and the thermodynamic limit.

In finite volume it is known that the spectrum of the transfer matrix is simple [67]. This means that if two states possess the same eigenvalue function $t(u)$ then they belong to the same $SU(2)$ multiplet. This also implies that if the spin quantum number $S_z$ is also specified, then the function $t(u)$ uniquely determines all Bethe roots. A practical procedure for recovering the Bethe roots from $t(u)$ is explained for example in [68].

Based on this, it might seem surprising, that the complete family $\{t_m(u)\}$ of TM's is needed in the $L \to \infty$ limit. Eq. (46) shows that the higher spin transfer matrices are algebraically dependent, and they can be expressed using the fundamental $t(u)$, thus the information stored in the complete family $\{t_m(u)\}$ might seem redundant.

The explanation for this apparent paradox is the following. Even though at finite $L$ the function $t(u)$ is enough the recover all Bethe roots, typically a large amount of information is lost by the thermodynamic limit. On a technical level this happens because for almost all $u$ one of the two terms in the expression (31) becomes dominant, and the other one becomes exponentially suppressed as $L \to \infty$. Thus it becomes impossible to reconstruct the root densities once the thermodynamic limit has been taken. However, further information is preserved in the other members of the family $\{t_m(u)\}$, such that eventually the set $\{X_m(u)\}$ remains complete in the TDL.

It is our goal to extend this picture to the higher rank cases. We will show that the situation is analogous to the $SU(2)$ case: the complete set of charges is obtained from the fusion hierarchy of the transfer matrices. However, before turning to the $SU(3)$ case we repeat some of the computations already present in the literature. We will use a slightly different approach, which is more convenient for later generalizations to the higher rank cases.

### 3.3 Inversion and quasi-locality

In our computations an important role will be played by certain asymptotic inversion relations. The main goal is to find some operators that invert the transfer matrices, such that the formal expressions $\partial_u \log(t_m(u)) = (t_m(u))^{-1} \partial_u t_m(u)$ can be made sense using well defined local objects. The transfer matrices themselves can not be inverted in the desired way, but there exist asymptotic inversion relations which hold in the $L \to \infty$ limit.

Such inversion relations are closely tied to the fusion of transfer matrices. Their study has a long history, which goes back to the seminal work of Baxter [58]. We do not attempt a thorough review of this topic, we merely mention a few references. For example, we will rely on some basic arguments about the inversion that already appeared in the work [69] of Pearce. Closely related ideas and methods appeared among others in [70, 71].

All of the asymptotic inversions that we will treat are based on a local inversion. In the case of the XXX model the Lax operators (43) satisfy the local inversion

$$\mathcal{L}^m(u)\mathcal{L}^m(-u) = \frac{-u^2 - (S_a \cdot \sigma_j + \frac{1}{2})^2}{-u^2 - (s + \frac{1}{2})^2} = 1. \tag{55}$$

This is most easily seen using the relation

$$S_a \cdot \sigma_j + \frac{1}{2} = \frac{(S_a + \sigma_j)^2 - (S_a)^2 - (\sigma_j)^2 + 1}{2}. \tag{56}$$

From the known values of the Casimir operators we compute the two possible eigenvalues of the operator in (56) as $\pm(s + 1/2)$, which implies the inversion (55).

Let us define a new family of transfer matrices, which are obtained simply by space reflection:

$$\bar{t}_m(u) = \mathrm{Tr}_a \left[ \mathcal{L}^m_{a,L}(u) \mathcal{L}^m_{a,L-1}(u) \ldots \mathcal{L}^m_{a,1}(u) \right]. \tag{57}$$

Using partial transpose in auxiliary space they can be expressed as

$$\bar{t}_m(u) = \mathrm{Tr}_a \left[ \left( \mathcal{L}^m_{a,1}(u) \right)^{t_a} \left( \mathcal{L}^m_{a,2}(u) \right)^{t_a} \ldots \left( \mathcal{L}^m_{a,L}(u) \right)^{t_a} \right]. \tag{58}$$

Furthermore, they can be related to the standard transfer matrices by a simple crossing transformation. All representations of $SU(2)$ are self-conjugate, therefore there exists a charge conjugation operator $C$ acting on the auxiliary space, such that it satisfies $C^2 = 1$ and performs conjugation as

$$\boldsymbol{S}^t = \boldsymbol{S}^* = -C\boldsymbol{S}C. \tag{59}$$

Therefore

$$\left( \mathcal{L}^m_{a,1}(u) \right)^{t_a} = C_a \frac{u + \mathrm{i}\frac{1}{2} - \mathrm{i}\boldsymbol{S}_a \cdot \boldsymbol{\sigma}_j}{u + \mathrm{i}\frac{m+1}{2}} C_a = \frac{u - \mathrm{i}\frac{m-1}{2}}{u + \mathrm{i}\frac{m+1}{2}} C_a \mathcal{L}^m_{a,1}(-u - \mathrm{i}) C_a. \tag{60}$$

The $C$-operators drop out when we compute the transfer matrices, we thus obtain the global crossing relation

$$\bar{t}_m(u) = \left( \frac{u - \mathrm{i}\frac{m-1}{2}}{u + \mathrm{i}\frac{m+1}{2}} \right)^L t_m(-u - \mathrm{i}). \tag{61}$$

In the following we will use the direct definition (57), because it is more advantageous for our purposes.

**Theorem 1.** *The following asymptotic inversion identity holds [25]:*

$$\bar{t}_m(-u) t_m(u) \approx 1, \qquad u \in \mathcal{P}. \tag{62}$$

*Here and in the following an asymptotic identity $A \approx B$ means that*

$$||A - B|| = \mathcal{O}(e^{-\alpha L}), \quad \alpha \in \mathbb{R}^+. \tag{63}$$

The above Theorem was treated in detail in [25]. We also sketch the proof using our conventions. First we shed some light on why the inversion relation holds.

Let us think about a different situation, and consider the monodromy matrices, i.e. the expressions (44)-(57) without the trace in auxiliary space:

$$\begin{aligned} M^m(u) &= \mathcal{L}^m_{a,1}(u) \mathcal{L}^m_{a,2}(u) \ldots \mathcal{L}^m_{a,L}(u), \\ \bar{M}^m(u) &= \mathcal{L}^m_{a,L}(u) \mathcal{L}^m_{a,L-1}(u) \ldots \mathcal{L}^m_{a,1}(u). \end{aligned} \tag{64}$$

In this case, the local inversion identity (55) immediately gives a global and exact inversion:

$$\bar{M}^m(-u) M^m(u) = 1. \tag{65}$$

The local steps leading to this global inversion are depicted in Fig. 2.

For the transfer matrices (62) the difficulty lies in the fact that the trace has been taken in auxiliary space. In this case we can not expect an exact inversion. Nevertheless, for certain values of $u$ and for large enough volumes the "boundary effect" of taking the trace does not propagate into the bulk of the chain. More precisely, it only causes an exponentially small effect.

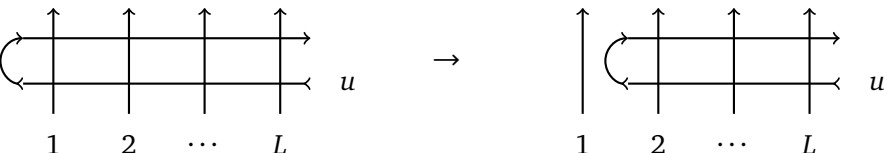

Figure 2: Graphical representation of the global inversion, which follows from consecutive local inversion steps. The product of monodromy matrices is equal to the identity after disentangling the local Lax operations.

A rigorous proof can be given by computing the norm of the difference

$$
\begin{aligned}
||\bar{t}_m(-u)t_m(u) - 1||^2 = {} & 2^{-L}\mathrm{Tr}\left((\bar{t}_m(-u)t_m(u))^\dagger \bar{t}_m(-u)t_m(u)\right) \\
& - 2^{-(L-1)}\Re\left[\mathrm{Tr}(\bar{t}_m(-u)t_m(u))\right] + 1.
\end{aligned}
\tag{66}
$$

The adjoints of the transfer matrices can be computed as

$$
(t_m(u))^\dagger = \bar{t}_m(-u^*).
\tag{67}
$$

Our aim is to show that

$$
\begin{aligned}
2^{-L}\mathrm{Tr}(\bar{t}_m(-u)t_m(u)) &\approx 1, \\
2^{-L}\mathrm{Tr}(t_m(u^*)\bar{t}_m(-u^*)\bar{t}_m(-u)t_m(u)) &\approx 1,
\end{aligned}
\tag{68}
$$

which will imply the asymptotic inversion. These two traces can be evaluated conveniently by building a corresponding 2D vertex model, see Figs. 2- 3. Here the action of the transfer matrices corresponds to adding a new row to the lattice, therefore we get two lattices of size $2 \times L$ and $4 \times L$. These partition functions can be evaluated in the "crossed channel" by building column-to-column transfer matrices. These are conventionally called Quantum Transfer Matrices (QTM's). The traces are evaluated using the eigenvalues of these QTM's. It follows from the local inversion relations, that the local delta-states given by $|\delta\rangle = \sum_j |j\rangle \otimes |j\rangle$ and $|\delta\rangle \otimes |\delta\rangle$ are eigenstates of the two-site and four-site QTM's, respectively. Their eigenvalues are simply 2, due to the local inversion and the trace over the physical space, see again Figs. 2-3. The identities in (68) are rigorously proven by showing that the delta-states are the dominant eigenstates. This requires a diagonalization of the QTM's in question. In the case of the two-site QTM this was performed in [25] using the $SU(2)$ algebra, whereas for the four-site case it was done analytically up to $s = 3/2$ and numerically for larger values of $s$.

We put forward that essentially the same steps are needed to prove the quasi-locality of the resulting charges. The reason for this is that based on the inversion we can write the asymptotic identity

$$
X_m(u) \approx (-\mathrm{i})\bar{t}_m(-u)\partial_u t_m(u),
\tag{69}
$$

and here the derivation $\partial_u$ acts only locally: the resulting operator will be translationally invariant and formally extensive. It remains to be shown that the HS norm of the traceless part scales linearly with the volume; this does not follow from (69), and only holds for $u \in \mathcal{P}$. For the computation of the HS norm one needs the same lattices of size $2 \times L$ and $4 \times L$ which were constructed above. The proof of quasi-locality follows relatively easily once the leading eigenvectors of the QTM's are found to be the delta states. This procedure is described in [25, 26], and we also explain it in Appendix A with the technical details in the case of the $SU(3)$-symmetric model.

We note that using the crossing relation (61) the asymptotic inversion (62) can be written in a form equivalent to the l.h.s. of the fusion relation (46) with a shifted rapidity. In this form

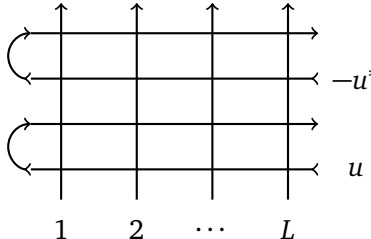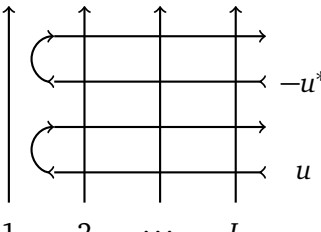

Figure 3: Graphical interpretation of the proof of the second equation in (68). The trace of the four transfer matrices in question is evaluated in the crossed channel by building a four-site Quantum Transfer Matrix, which adds one column to the diagram. A specific eigenvector of this QTM is the product of two delta-states: this state has eigenvalue 2, which follows from the local inversion as depicted in the graph, after taking the trace in the vertical direction. The desired identity in (68) holds if this is the leading eigenvalue.

the asymptotic inversion states that on the r.h.s. the scalar (state independent) part will be dominant in the thermodynamic limit.

The inversion relation has important consequences for the transfer matrix eigenvalues. Eq. (62) has to hold on the level of eigenvalues for *almost all* states. We now analyze the implications of this, by computing the products $\bar{t}_m(-u)t_m(u)$.

The eigenvalues of the space reflected TM could be computed from the crossing relation (61), but there is a more direct way. It is known that in Bethe Ansatz space reflection can be represented by the change $\{\lambda\}_N \to \{-\lambda\}_N$. The TM eigenvalues involve ratios of $Q$-functions with certain shifts. Changing the sign of both $u$ and all the rapidities is equivalent to changing the signs of all the shift parameters. Thus the eigenvalues of $\bar{t}_m(-u)$ are immediately found from (48):

$$\bar{t}_m(-u) = \frac{Q_1\left(u + i\frac{m}{2}\right)Q_1\left(u - i\frac{m+2}{2}\right)}{Q_0\left(u - i\frac{m+1}{2}\right)} \sum_{k=0}^{m} \frac{Q_0\left(u - i\frac{m+1}{2} + ik\right)}{Q_1\left(u - i\frac{m}{2} + ik\right)Q_1\left(u - i\frac{m+2}{2} + ik\right)}. \quad (70)$$

Let us now consider the product of the eigenvalues $\bar{t}_m(-u)t_m(u)$. Expanding the product we obtain a sum of $(m+1)^2$ terms, each of which involves ratios of $Q$-functions. Inspection shows that among these $(m+1)^2$ terms there will be a single one which gives identically 1; this will come from the summands with index $k = 0$. All the remaining terms are ratios of $Q$-functions, which are exponentially increasing or decreasing with $L$, depending on $u$. It follows from the inversion identity that if $u$ is within the physical strip, then all of these terms have to be exponentially decreasing. This also implies, that from the $(m+1)$ terms in the eigenvalues of $t_m(u)$ the $k = 0$ term has to be the leading one for almost all states if $u \in \mathcal{P}$. We can thus write the explicit formula

$$t_m(u) \approx \frac{Q_1\left(u + i\frac{m}{2}\right)}{Q_1\left(u - i\frac{m}{2}\right)}, \qquad u \in \mathcal{P}, \quad m = 1, 2, \ldots \quad (71)$$

These relations play an essential role in establishing the string-charge identities (51), see the original papers [24, 46].

We will show that similar steps are needed also in the $SU(3)$-symmetric model. That model has a more complicated Bethe Ansatz solution and corresponding fusion hierarchy of transfer matrices, nevertheless the inversion relations take an identical form, and are equally important for the derivations of the string-charge relations.

# 4 The $SU(3)$-symmetric Lai-Sutherland model

In the $SU(3)$ symmetric Lai-Sutherland model the eigenstates are constructed as excitations over the reference state, which can be chosen for example as $|\emptyset\rangle = |1\rangle \otimes \cdots \otimes |1\rangle$, where $\{|1\rangle, |2\rangle, |3\rangle\}$ is an orthonormal basis of $\mathbb{C}^3$. One-particle excitations are spin waves which carry an internal degree of freedom corresponding to the polarization, thus carrying the defining representation of $SU(2)$. General multi-particle excited states can be considered as interacting spin waves, and each state is described by the momenta of the particles and also by an auxiliary wave function describing the orientation in the resulting internal space.

Corresponding to this physical picture, the Bethe states of this model can be characterized by two sets of rapidity parameters $\{\lambda_j\}_{j=1}^N$ and $\{\mu_j\}_{j=1}^M$. Here $N$ is the number of physical particles and the $\lambda_j$ describe their quasi-momenta. Further, the secondary (or magnonic) rapidities $\{\mu_j\}_{j=1}^M$ describe the orientation in the internal space; they can be understood as the Bethe rapidities of an auxiliary spin chain problem. The $GL(3)$ global quantum numbers are $(L-N, N-M, M)$, and it is required that $N \le 2L/3$ and $M \le N/2$.

The un-normalized real space wave functions can be written as

$$|\boldsymbol{\lambda}_N, \boldsymbol{\mu}_M\rangle = \sum_{1 \le n_1 < \ldots < n_N \le L} \sum_{1 \le m_1 < \ldots < m_M \le N} \sum_{\mathcal{P} \in \mathcal{S}^N} \left( \prod_{1 \le r < l \le N} \frac{\lambda_{\mathcal{P}(l)} - \lambda_{\mathcal{P}(r)} - \mathrm{i}}{\lambda_{\mathcal{P}(l)} - \lambda_{\mathcal{P}(r)}} \right)$$

$$\times \langle \boldsymbol{m} | \boldsymbol{\lambda}_{\mathcal{P}}, \boldsymbol{\mu} \rangle \prod_{r=1}^N \left( \frac{\lambda_{\mathcal{P}(r)} + \frac{\mathrm{i}}{2}}{\lambda_{\mathcal{P}(r)} - \frac{\mathrm{i}}{2}} \right)^{n_r} \prod_{r=1}^M (E_{32})_{m_r} \prod_{s=1}^N (E_{21})_{n_s} |\emptyset\rangle,$$

where we used the elementary matrices $E_{ji} = |j\rangle\langle i|$. The wave function amplitudes are given by

$$\langle \boldsymbol{m} | \boldsymbol{\lambda}_{\mathcal{P}}, \boldsymbol{\mu} \rangle = \sum_{\mathcal{R} \in \mathcal{S}^M} A(\boldsymbol{\lambda}_{\mathcal{R}}) \prod_{\ell=1}^M F_{\boldsymbol{\lambda}_{\mathcal{P}}}(\mu_{\mathcal{R}(\ell)}; m_\ell),$$

$$F_{\boldsymbol{\lambda}}(\mu, s) = \frac{-\mathrm{i}}{\mu - \lambda_s - \frac{\mathrm{i}}{2}} \prod_{n=1}^{s-1} \frac{\mu - \lambda_n + \frac{\mathrm{i}}{2}}{\mu - \lambda_n - \frac{\mathrm{i}}{2}}, \tag{72}$$

$$A(\boldsymbol{\lambda}) = \prod_{1 \le r < l \le M} \frac{\mu_l - \mu_r - \mathrm{i}}{\mu_l - \mu_r}.$$

It follows from the periodicity of the wave function that the two sets of rapidities are solutions to the following Bethe equations:

$$\left( \frac{\lambda_j + \frac{\mathrm{i}}{2}}{\lambda_j - \frac{\mathrm{i}}{2}} \right)^L = \prod_{k=1, k \ne j}^N \frac{\lambda_j - \lambda_k + \mathrm{i}}{\lambda_j - \lambda_k - \mathrm{i}} \prod_{k=1}^M \frac{\lambda_j - \mu_k - \frac{\mathrm{i}}{2}}{\lambda_j - \mu_k + \frac{\mathrm{i}}{2}}, \qquad j = 1, \ldots, N$$

$$1 = \prod_{k=1}^M \frac{\mu_j - \lambda_k - \frac{\mathrm{i}}{2}}{\mu_j - \lambda_k + \frac{\mathrm{i}}{2}} \prod_{k=1, k \ne j}^N \frac{\mu_j - \mu_k + \mathrm{i}}{\mu_j - \mu_k - \mathrm{i}}, \qquad j = 1, \ldots, M. \tag{73}$$

The lattice momentum and the energy of a Bethe state $\left| \{\lambda_j\}_{j=1}^N, \{\mu_j\}_{j=1}^M \right\rangle$ is given by the sum of one particle momentum and energy, respectively:

$$P = \sum_{j=1}^N p(\lambda_j), \qquad\qquad p(\lambda) = \mathrm{i} \log \left( \frac{\lambda_j + \frac{\mathrm{i}}{2}}{\lambda_j - \frac{\mathrm{i}}{2}} \right), \tag{74}$$

$$E = \sum_{j=1}^N \varepsilon(\lambda_j), \qquad\qquad \varepsilon(\lambda) = -\frac{1}{\lambda^2 + \frac{1}{4}}. \tag{75}$$

Note that both the momentum and the energy depend only on the Bethe roots of the first type.

The eigenvalues of the fundamental transfer matrices (15) on Bethe states can be computed as [57]

$$t(u) = \frac{Q_1\left(u - \frac{i}{2}\right)}{Q_1\left(u + \frac{i}{2}\right)} + \frac{Q_0(u)Q_1\left(u + \frac{3i}{2}\right)Q_2(u)}{Q_0(u+i)Q_1\left(u + \frac{i}{2}\right)Q_2(u+i)} + \frac{Q_0(u)Q_2\left(u + 2i\right)}{Q_0(u+i)Q_2\left(u+i\right)}, \tag{76}$$

where

$$Q_0(u) = u^L \tag{77}$$

and

$$Q_1(u) = \prod_{j=1}^{N}(u - \lambda_j), \qquad Q_2(u) = \prod_{j=1}^{M}(u - \mu_j) \tag{78}$$

are the $Q$-functions associated with the first and second level Bethe roots.

The eigenvalues of the local charges $\mathcal{Q}_k$ can be computed from (76) using the definition (17). It follows from the presence of the $Q_0(u)$ factor in the second and third terms of (76) that only the first term will contribute to the eigenvalue of $\mathcal{Q}_k$ as long as $k < L$. This implies that the local charges only depend on the first level Bethe roots $\{\lambda_j\}_{j=1}^{N}$. This already indicates that the local charges cannot give a full description of the states. Note that (75) can be derived immediately from (76).

## 4.1 String hypothesis and Thermodynamic Bethe Ansatz

In the Lai-Sutherland model both the first and second type of rapidities can form strings, and they form the same patterns in the complex plain. A Bethe state with $M_n^{(1)}$ and $M_n^{(2)}$ number of $n$-strings for the first and second type of particles thus consists of the rapidities

$$\begin{aligned}
\lambda_\alpha^{n,\ell} &= \lambda_\alpha^n + i\left(\frac{n+1}{2} - \ell\right) + \delta_{1,\alpha}^{n,\ell}, \quad \ell = 1,\dots,n, \quad \alpha = 1,\dots,M_n^{(1)}, \\
\mu_\alpha^{n,\ell} &= \mu_\alpha^n + i\left(\frac{n+1}{2} - \ell\right) + \delta_{2,\alpha}^{n,\ell}, \quad \ell = 1,\dots,n, \quad \alpha = 1,\dots,M_n^{(2)}.
\end{aligned} \tag{79}$$

Here, $\lambda_\alpha^n$, $\mu_\alpha^n \in \mathbb{R}$ are the string centers, and $\delta_{1,\alpha}^{n,\ell}$, $\delta_{2,\alpha}^{n,\ell}$ are the string deviations, exponentially small in large volumes. The total number of Bethe roots is computed simply as

$$\sum_{n=1}^{\infty} n M_n^{(1)} = N, \qquad \sum_{n=1}^{\infty} n M_n^{(2)} = M. \tag{80}$$

We will be interested in the thermodynamic limit, when $N \to \infty$, $M \to \infty$, $L \to \infty$ while we keep the ratios $N/L$ and $M/L$ fixed. The string hypothesis states that only Bethe roots in the form of (79) contribute to thermodynamic behaviour. For the string centers we introduce the densities $\rho_n^{(1)}(u)$, $\rho_n^{(2)}(u)$, which are normalized such that in a large volume $L$ the total number of $n$-strings of the first/second type between rapidities $u$ and $u + \Delta u$ is $L\rho_n^{(1/2)}(u)\Delta u$. Similarly we introduce the hole densities $\rho_{h,n}^{(1)}, \rho_{h,m}^{(2)}$.

In the thermodynamic limit the Bethe equations (73) can be transformed into a set of coupled linear integral equations:

$$\begin{aligned}
\rho_{t,n}^{(1)}(\lambda) &= a_n(\lambda) - \sum_{m=1}^{\infty} a_{n,m} \star \rho_m^{(1)}(\lambda) + \sum_{m=1}^{\infty} b_{n,m} \star \rho_m^{(2)}(\lambda), \\
\rho_{t,n}^{(2)}(\lambda) &= -\sum_{m=1}^{\infty} a_{n,m} \star \rho_m^{(2)}(\lambda) + \sum_{m=1}^{\infty} b_{n,m} \star \rho_m^{(1)}(\lambda),
\end{aligned} \tag{81}$$

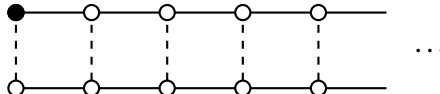

Figure 4: TBA diagram of the $SU(3)$-symmetric model. The node in row $a$ and column $m$ from the top left correspond to the $m$-string of particle type $a = 1, 2$. The links denote the two different convolutions in Eq. (84). The filling of the top left node signals the source term for $\rho_{t,1}^{(1)}(u)$.

where the total root densities are

$$\rho_{t,n}^{(j)}(u) = \rho_n^{(j)}(u) + \rho_{h,n}^{(j)}(u), \qquad j = 1, 2. \tag{82}$$

The kernels are

$$
\begin{aligned}
a_{n,m}(\lambda) &= (1 - \delta_{nm})a_{|n-m|}(\lambda) + 2a_{|n-m|+2}(\lambda) + \dots 2a_{n+m-2}(\lambda) + a_{n+m}(\lambda), \\
b_{n,m}(\lambda) &= a_{|n-m|+1}(\lambda) + 2a_{|n-m|+3}(\lambda) + \dots 2a_{n+m-1}(\lambda),
\end{aligned} \tag{83}
$$

and $a_n(\lambda)$ is defined in (52).

Similarly to the spin-1/2 case, the aforementioned equations can be cast in a partially decoupled form (for derivation, see e.g. [36]):

$$
\begin{aligned}
\rho_{t,n}^{(1)}(\lambda) &= \delta_{n,1}s(\lambda) + s \star \left( \rho_{h,n-1}^{(1)} + \rho_{h,n+1}^{(1)} \right)(\lambda) + s \star \rho_n^{(2)}(\lambda), \\
\rho_{t,n}^{(2)}(\lambda) &= s \star \left( \rho_{h,n-1}^{(2)} + \rho_{h,n+1}^{(2)} \right)(\lambda) + s \star \rho_n^{(1)}(\lambda),
\end{aligned} \tag{84}
$$

where the following definitions and conventions are understood:

$$\rho_{h,0}^{(r)}(u) = 0, \quad r = 1, 2, \tag{85}$$

and $s(u)$ is defined in (39). The structure of these integral equations is depicted on the "TBA diagram" 4.

## 4.2 Strategy towards the string-charge relations

It is our goal to find a set of quasi-local charges, which will uniquely determine all root densities $\rho_m^{(1)}(u)$ and $\rho_m^{(2)}(u)$. Based on the XXX chain it is a natural idea to consider the fusion hierarchy of the transfer matrices.

In the $SU(3)$-symmetric model there is a set of fused transfer matrices $t_m^{(a)}(u)$ with $a = 1, 2$, and $m = 1, 2, \dots, \infty$. They correspond to representations of $SU(3)$ described by the rectangular Young diagram with $a$ rows and $m$ columns. Precise definitions of $t_m^{(a)}(u)$ using local Lax operators will be given in Section 6. We put forward that these TM's satisfy the Hirota equation or $T$-system [62, 63]

$$
t_m^{(a)}\left(u + \frac{i}{2}\right) t_m^{(a)}\left(u - \frac{i}{2}\right) = t_{m+1}^{(a)}(u)t_{m-1}^{(a)}(u) + t_m^{(a-1)}(u)t_m^{(a+1)}(u), \tag{86}
$$
$$a = 1, 2, \quad m = 1, 2, \dots$$

Here $t_0^{(a)}(u) = 1$ by definition and $t_m^{(0)}(u)$ and $t_m^{(3)}(u)$ are scalar functions that will be specified later.

The structure of this $T$-system is very closely related to the "TBA diagram" 4. The top and bottom rows correspond to the transfer matrices with $a = 1$ and $a = 2$, respectively. Based

on previous experience we expect to find a relation between the eigenvalues of these fused transfer matrices and the Bethe root densities. We put forward the formal definition

$$X_m^{(a)}(u) = (-\mathrm{i}) \frac{\partial}{\partial u} \log t_m^{(a)}(u). \tag{87}$$

We will show that the $X_m^{(a)}(u)$ can be constructed locally, and their mean values are connected to the root densities by relations analogous to (54). Furthermore, we will prove in two specific cases that they are quasi-local within the physical strip.

# 5 Quasi-local charges: the defining and conjugate representations

In this section we investigate two special transfer matrices in the $SU(3)$-case: the operators $t_1^{(1)}(u)$ and $t_1^{(2)}(u)$ that correspond to the defining and conjugate representations of $SU(3)$, respectively.

The first one is the fundamental TM, corresponding to the defining representation with highest weight $(1, 0)$ and the Young diagram with one box. For this TM we will use the notation $t^3(u)$, and the definition is the same as in (15):

$$t^3(u) = \mathrm{Tr}_0 R_{10}(u) R_{20}(u) \dots R_{L0}(u), \tag{88}$$

where $R(u)$ is given by (11) with $N = 3$.

The second special TM corresponds to the conjugate (or anti-symmetric tensor) representation of $SU(3)$, given by highest weight $(0, 1)$ and the Young diagram with two rows and one column. We use the notation $t^{\bar{3}}(u)$ and the definition

$$t^{\bar{3}}(u) = \mathrm{Tr}_0 R_{10}^{\bar{3}}(u) R_{20}^{\bar{3}}(u) \dots R_{L0}^{\bar{3}}(u), \tag{89}$$

where $R^{\bar{3}}(u)$ is the $R$-matrix acting on the tensor product of a fundamental and conjugate representation. It is given by

$$R^{3,\bar{3}}(u) \equiv R^{\bar{3}}(u) = \frac{u + \frac{3\mathrm{i}}{2} - \mathrm{i}K}{u + \frac{3\mathrm{i}}{2}}. \tag{90}$$

Here $K$ is the trace operator (or Temperley-Lieb operator) with matrix elements $K_{ab}^{cd} = \delta_{ab}\delta_{cd}$. Note that $K$ is the partial transpose of the permutation operator: $P^{t_1} = P^{t_2} = K$. Therefore we have the simple relation

$$R^{\bar{3}}(u) = \frac{u + \frac{\mathrm{i}}{2}}{u + \frac{3\mathrm{i}}{2}} R^{t_1}\left(-u - \frac{3\mathrm{i}}{2}\right). \tag{91}$$

This $R$-matrix also satisfies the unitarity relation:

$$R^{\bar{3}}(u) R^{\bar{3}}(-u) = 1, \tag{92}$$

which is easily checked using the Temperley-Lieb property $K^2 = 3K$.

For this special conjugate $R$-matrix the compatibility conditions are easily derived from the original Yang-Baxter relation (12). Making use of (91) and (92) it is easy to show that

$$R_{23}(v - w) \bar{R}_{13}(u - w) \bar{R}_{12}(u - v) = \bar{R}_{12}(u - v) \bar{R}_{13}(u - w) R_{23}(v - w). \tag{93}$$

It follows that the TM's $\{t_m^3(u), t_m^{\bar{3}}(u)\}$ are all commuting with each other.

The group invariance property of the conjugate transfer matrix follows easily from (14). Taking a partial transpose in the second space we get

$$G_1(G_2^t)^{-1}R^{t_2}(u) = R^{t_2}(u)G_1(G_2^t)^{-1}, \qquad G \in GL(3). \tag{94}$$

For any $G \in SU(3)$ we have $(G^t)^{-1} = G^*$, thus we see by the relation (91) that the matrix $R^{\bar{3}}(u)$ is compatible with representations $3 \otimes \bar{3}$.

Similar to the case of the XXX chain it is very useful to define the space reflected transfer matrices corresponding to these two representations:

$$\bar{t}^3(u) = \text{Tr}_0 R^3_{L0}(u) \dots R^3_{10}(u), \tag{95}$$

$$\bar{t}^{\bar{3}}(u) = \text{Tr}_0 R^{\bar{3}}_{L0}(u) \dots R^{\bar{3}}_{10}(u). \tag{96}$$

The local crossing relation 91 implies the following connections:

$$\bar{t}^{\bar{3}}(u) = \left(\frac{u+\frac{i}{2}}{u+\frac{3i}{2}}\right)^L t^3\left(-u-\frac{3i}{2}\right),$$

$$\bar{t}^3(u) = \left(\frac{u}{u+i}\right)^L t^{\bar{3}}\left(-u-\frac{3i}{2}\right). \tag{97}$$

These are simple generalizations of the crossing relation (61) in the $SU(2)$ case. Even though the space reflected operators are not independent, we keep their definition and this special notation, because they are very useful to study the inversion relations and quasi-locality properties of the charges.

For example, the adjoints of the transfer matrices can be expressed simply as

$$(t^3)^\dagger(u) = \bar{t}^{\bar{3}}(-u^*), \qquad (t^{\bar{3}}(u))^\dagger = \bar{t}^3(-u^*). \tag{98}$$

For the details see Lemma 1 in the Appendix.

**Theorem 2.** *For $t^3(u)$ and $t^{\bar{3}}(u)$ the following asymptotic inversion relations hold when $u \in \mathcal{P}$*

$$\bar{t}^3(-u)t^3(u) \approx 1,$$

$$\bar{t}^{\bar{3}}(-u)t^{\bar{3}}(u) \approx 1. \tag{99}$$

A rigorous proof is presented in Appendix A. The proof is based on the same ideas as explained in the case of the XXX model. The core relations are the local inversions (13)-(92), which guarantee the exact inversion of the *monodromy matrices*. Considering the transfer matrices, the "boundary effect" of taking the trace does not propagate into the bulk of the chain, if $u$ is chosen from the physical strip.

Based on the results of the $SU(2)$-symmetric chain we define the following two generating functions for the charges:

$$X(u) = (-i)\partial_u \log t(u),$$

$$Y(u) = (-i)\partial_u \log t^{\bar{3}}(u). \tag{100}$$

Due to the asymptotic inversion they can be written in large enough volumes as

$$X(u) \approx (-i)\bar{t}(-u)\partial t(u),$$

$$Y(u) \approx (-i)\bar{t}^{\bar{3}}(-u)\partial t^{\bar{3}}(u). \tag{101}$$

The formula (100) is convenient for the treatment of the eigenvalues, whereas the advantage of (101) is its local construction, enabling the evaluation of mean values in initial states and the proof of quasi-locality.

It follows from the asymptotic inversion and the adjoint property (98) that for $u \in \mathbb{R}$ these operators are Hermitian.

**Theorem 3.** *The traceless operators $\{X(u)\}$ and $\{Y(u)\}$ are quasi-local within the physical strip.*

The detailed proof of this Theorem is presented in appendix A. The proof uses the same techniques as in the original works [25, 26], and the starting point is the existence of local inversion relations for the *R*-matrices.

Let us now investigate the consequences of the inversion relation for the eigenvalues of these transfer matrices.

For the fundamental TM and its space reflected counterpart we have

$$
\begin{aligned}
t^3(u) &= \frac{Q_1\left(u-\frac{i}{2}\right)}{Q_1\left(u+\frac{i}{2}\right)} + \frac{Q_0(u)Q_1\left(u+\frac{3i}{2}\right)Q_2(u)}{Q_0(u+i)Q_1\left(u+\frac{i}{2}\right)Q_2(u+i)} + \frac{Q_0(u)Q_2\left(u+2i\right)}{Q_0(u+i)Q_2\left(u+i\right)}, \\
\bar{t}^3(-u) &= \frac{Q_1\left(u+\frac{i}{2}\right)}{Q_1\left(u-\frac{i}{2}\right)} + \frac{Q_0(u)Q_1\left(u-\frac{3i}{2}\right)Q_2(u)}{Q_0(u-i)Q_1\left(u-\frac{i}{2}\right)Q_2(u-i)} + \frac{Q_0(u)Q_2\left(u-2i\right)}{Q_0(u-i)Q_2\left(u-i\right)},
\end{aligned}
\tag{102}
$$

where the second equality follows simply from the fact that in Bethe Ansatz the space reflection is described by negating all Bethe rapidities.

Using the relations (97) we obtain further

$$
\begin{aligned}
t^{\bar{3}}(u) &= \frac{Q_0\left(u+\frac{i}{2}\right)Q_1\left(u+2i\right)}{Q_0\left(u+\frac{3i}{2}\right)Q_1\left(u+i\right)} + \frac{Q_1(u)Q_2\left(u+\frac{3i}{2}\right)}{Q_1(u+i)Q_2\left(u+\frac{i}{2}\right)} + \frac{Q_2\left(u-\frac{i}{2}\right)}{Q_2\left(u+\frac{i}{2}\right)}, \\
\bar{t}^{\bar{3}}(-u) &= \frac{Q_0\left(u-\frac{i}{2}\right)Q_1\left(u-2i\right)}{Q_0\left(u-\frac{3i}{2}\right)Q_1\left(u-i\right)} + \frac{Q_1(u)Q_2\left(u-\frac{3i}{2}\right)}{Q_1(u-i)Q_2\left(u-\frac{i}{2}\right)} + \frac{Q_2\left(u+\frac{i}{2}\right)}{Q_2\left(u-\frac{i}{2}\right)}.
\end{aligned}
\tag{103}
$$

Let us now investigate the inversion relations (99) on the level of these eigenvalues. Multiplying the sums we observe that in both cases there will be 9 terms out which only one is identically equal to 1. The remaining 8 terms will include various ratios of *Q*-functions. In the large volume limit these ratios will be exponentially increasing or decreasing, depending on $u$. It follows from the inversion relation, that within the individual sums those terms have to be dominant for $u \in \mathcal{P}$ which produce the required identity. We thus have the relations for $u \in \mathcal{P}$

$$
\begin{aligned}
t^3(u) &\approx \frac{Q_1\left(u-\frac{i}{2}\right)}{Q_1\left(u+\frac{i}{2}\right)}, \\
t^{\bar{3}}(u) &\approx \frac{Q_2\left(u-\frac{i}{2}\right)}{Q_2\left(u+\frac{i}{2}\right)}.
\end{aligned}
\tag{104}
$$

These are crucial in establishing the thermodynamic limit of the charges and eventually the string-charge relations.

Notice the symmetry of these relations: exchanging the defining and conjugate representations simply corresponds to exchanging the two types of Bethe rapidities. This is simply the conjugation symmetry of the Dynkin diagram of $SU(3)$, which is nicely reflected by the Bethe Ansatz solution (compare with Fig 4).

It is important that (104) does not necessarily hold for *all* Bethe states. For example, if we choose $u \in \mathbb{R}$ and keep the number of rapidities finite while performing the $L \to \infty$ limit then there will be two remaining finite terms in $t^{\bar{3}}(u)$. Our proof using the inversion relation only tells us that (104) will hold for almost all states, where the probability measure is derived simply from the HS scalar product. This corresponds to the infinite temperature thermal ensemble. Therefore, in the first instance our statement only concerns the infinite temperature Bethe states. Nevertheless, it can be argued based on continuity that (104) still holds for Bethe root densities "close" to the infinite temperature state. We give further comments on this issue at the end of the next Section.

# 6 Arbitrary representations and string-charge relations

In this section we treat all the representations of $SU(3)$ that correspond to rectangular Young diagrams. To this order we define the families $t_m^3(u) \equiv t_m^{(1)}(u)$ and $t_m^{\bar{3}}(u) \equiv t_m^{(2)}(u)$ with $m = 1, 2, \dots$. For $m = 1$ they coincide with the two transfer matrices $t^3(u)$ and $t^{\bar{3}}(u)$ of the previous Section.

For a representation of $\Lambda$ of $GL(N)$ with rectangular Young diagrams the $R$-matrix acting on the tensor product of the defining representation and $\Lambda$ can be expressed as [60, 72, 73]

$$R^\Lambda(u) \quad \sim \quad u + \mathrm{i} \sum_{i,j} E_{ij} \Lambda_{ji}, \tag{105}$$

where $E_{ij}$ are the elementary matrices acting on the defining representation, $\Lambda_{ij}$ are their representations, and there is an arbitrary normalization factor not included in (105). If $\Lambda$ is the defining representation, we get back the usual $R$-matrix, as $P = \sum_{i,j} E_{ij} E_{ji}$.

Let $\Lambda_{ij}^{(m,0)}$ and $\Lambda_{ij}^{(0,m)}$ be the representation matrices of $GL(3)$ corresponding to the $1 \times m$ Young diagram with $GL(3)$ highest weight $(m, 0, 0)$ and to the $2 \times m$ Young diagram with $GL(3)$ highest weight $(m, m, 0)$, respectively. For these two families of representations we introduce the normalized $R$-matrices

$$\begin{aligned}
R^{(m,0)}(u) &= \frac{u - \mathrm{i}\frac{m-1}{2} + \mathrm{i} \sum_{i,j} E_{ij} \Lambda_{ji}^{(m,0)}}{u + \mathrm{i}\frac{m+1}{2}}, \\
R^{(0,m)}(u) &= \frac{u - \mathrm{i}\frac{m-2}{2} + \mathrm{i} \sum_{i,j} E_{ij} \Lambda_{ji}^{(0,m)}}{u + \mathrm{i}\frac{m+2}{2}}.
\end{aligned} \tag{106}$$

Compared to (105) they involve a scalar factor and a shift in the rapidity, which is equivalent to adding a constant to all components of the weight vector.

These $R$-matrices satisfy the local inversion relations

$$R^{(m,0)}(u) R^{(m,0)}(-u) = 1, \qquad R^{(0,m)}(u) R^{(0,m)}(-u) = 1. \tag{107}$$

This is a general property of $R$-matrices [72, 73], but we also prove it explicitly in Appendix D.

Now we define the transfer matrices

$$\begin{aligned}
t_m^3(u) &= \mathrm{Tr}_a R_{1,a}^{(m,0)}(u) R_{2,a}^{(m,0)}(u) \dots R_{L,a}^{(m,0)}(u), \\
t_m^{\bar{3}}(u) &= \mathrm{Tr}_a R_{1,a}^{(0,m)}(u) R_{2,a}^{(0,m)}(u) \dots R_{L,a}^{(0,m)}(u).
\end{aligned} \tag{108}$$

In analogy with the previous Sections we also define the space reflected transfer matrices

$$\begin{aligned}
\bar{t}_m^3(u) &= \mathrm{Tr}_a R_{L,a}^{(m,0)}(u) R_{L-1,a}^{(m,0)}(u) \dots R_{1,a}^{(m,0)}(u), \\
\bar{t}_m^{\bar{3}}(u) &= \mathrm{Tr}_a R_{L,a}^{(0,m)}(u) R_{L-1,a}^{(0,m)}(u) \dots R_{1,a}^{(0,m)}(u).
\end{aligned} \tag{109}$$

They are not independent from the TM's in (108), they are related by crossing transformations of the type (97). However, these relations are not relevant for our purposes.

Based on our earlier results we formulate the following:

**Conjecture 1.** *For $u \in \mathcal{P}$ the following asymptotic inversion relations hold:*

$$\begin{aligned}
\bar{t}_m^3(-u) t_m^3(u) &\approx 1, \\
\bar{t}_m^{\bar{3}}(-u) t_m^{\bar{3}}(u) &\approx 1.
\end{aligned} \tag{110}$$

The main idea behind this is the same as earlier: the global inversion of *monodromy* matrices follows from the local inversion relations (107). Then the asymptotic inversion of transfer matrices will hold for $u \in \mathcal{P}$ if the "boundary effect" of taking the trace does not propagate into the bulk of the chain. This could be checked by constructing the appropriate 2-site and 4-site Quantum Transfer Matrices, in analogy with the computations of the previous Sections. At present we do not have a general proof, except for the two special cases with $m = 1$, presented above and in Appendix A.

We define the following two families of generating functions for quasi-local charges:

$$
\begin{aligned}
X_m(u) &= (-\mathrm{i})\partial_u \log t_m^3(u), \\
Y_m(u) &= (-\mathrm{i})\partial_u \log t_m^{\bar{3}}(u).
\end{aligned}
\tag{111}
$$

Under the assumption of the above conjecture, they are asymptotically equal to the following locally constructed quantities:

$$
\begin{aligned}
X_m(u) &\approx (-\mathrm{i})t_m^3(-u)(\partial\, t_m^3)(u), \\
Y_m(u) &\approx (-\mathrm{i})t_m^{\bar{3}}(-u)(\partial\, t_m^{\bar{3}})(u).
\end{aligned}
\tag{112}
$$

**Conjecture 2.** *For $u \in \mathcal{P}$ the traceless operators $\{X_m(u)\}$ and $\{Y_m(u)\}$ are quasi-local.*

We do not have a proof for this conjecture, but the cases of $\{X_1(u)\}$ and $\{Y_1(u)\}$ we treated in the previous section, together with the known results for the $SU(2)$ case already give strong motivation for its validity.

Let us now treat the eigenvalues of the transfer matrices and the charges. The eigenvalues of $t_m^3(u)$ and $t_m^{\bar{3}}(u)$ can be expressed using the $Q$-functions as

$$
\begin{aligned}
t_m^3(u) &= \frac{Q_1\left(u - \mathrm{i}\frac{m}{2}\right)Q_2\left(u + \mathrm{i}\frac{m+3}{2}\right)}{Q_0\left(u + \mathrm{i}\frac{m+1}{2}\right)} \sum_{k=0}^{m} \frac{Q_0\left(u + \mathrm{i}\frac{m+1}{2} - \mathrm{i}k\right)Q_2\left(u + \mathrm{i}\frac{m+1}{2} - \mathrm{i}k\right)}{Q_1\left(u + \mathrm{i}\frac{m}{2} - \mathrm{i}k\right)Q_1\left(u + \mathrm{i}\frac{m+2}{2} - \mathrm{i}k\right)} \times \\
&\qquad \times \sum_{\ell=0}^{k} \frac{Q_1\left(u + \mathrm{i}\frac{m+2}{2} - \mathrm{i}\ell\right)}{Q_2\left(u + \mathrm{i}\frac{m+1}{2} - \mathrm{i}\ell\right)Q_2\left(u + \mathrm{i}\frac{m+3}{2} - \mathrm{i}\ell\right)}, \\
t_m^{\bar{3}}(u) &= \frac{Q_1\left(u + \mathrm{i}\frac{m+3}{2}\right)Q_2\left(u - \mathrm{i}\frac{m}{2}\right)}{Q_0\left(u + \mathrm{i}\frac{m+2}{2}\right)} \sum_{k=0}^{m} \frac{Q_0\left(u - \mathrm{i}\frac{m-2}{2} + \mathrm{i}k\right)Q_2\left(u - \mathrm{i}\frac{m-2}{2} + \mathrm{i}k\right)}{Q_1\left(u - \mathrm{i}\frac{m-3}{2} + \mathrm{i}k\right)Q_1\left(u - \mathrm{i}\frac{m-1}{2} + \mathrm{i}k\right)} \times \\
&\qquad \times \sum_{\ell=0}^{k} \frac{Q_1\left(u - \mathrm{i}\frac{m-1}{2} + \mathrm{i}\ell\right)}{Q_2\left(u - \mathrm{i}\frac{m}{2} + \mathrm{i}\ell\right)Q_2\left(u - \mathrm{i}\frac{m-2}{2} + \mathrm{i}\ell\right)}.
\end{aligned}
\tag{113}
$$

These explicit formulas can be derived from the more general "tableaux sum" valid in the $SU(N)$-symmetric model [74]. The concrete formula for the general "tableaux sum" will be given in the next Section.

It can be checked by direct substitution that these eigenvalue functions solve the Hirota equation (86) with boundary conditions $t_0^{(a)} = 1$ and

$$
t_m^{(0)}(u) = \frac{Q_0(u - \mathrm{i}\frac{m}{2})}{Q_0(u + \mathrm{i}\frac{m}{2})}, \qquad t_m^{(3)}(u) = 1.
\tag{114}
$$

These boundary conditions might seem somewhat unnatural: they differ from the most often used conventions, see for example the comparison on page 18. of [62]. We apply this normalization so that the inversion relations hold without additional "kinematical" $Q_0$ factors. Also, this normalization is most convenient for the string-charge relations.

The matching of the formulas (113) with the normalization of the Lax matrices (106) is checked easily by computing the eigenvalues on the reference state $|\emptyset\rangle = |111\ldots1\rangle$ . This

amounts to setting $Q_1(u) = Q_2(u) = 1$, and comparing the remaining ratios of $Q_0(u)$ functions to the direct application of (106) using the eigenvalues of the representation matrices $\Lambda_{11}^{(m,0)}$ and $\Lambda_{11}^{(0,m)}$.

In order to derive the string-charge relations, we are interested in the thermodynamic limit of the above eigenvalues. We will argue that in the thermodynamic limit for $u \in \mathcal{P}$ the leading terms in the above sums are:

$$t_m^3(u) \approx \lim_{\text{TDL}} \frac{Q_1\left(u - i\frac{m}{2}\right)}{Q_1\left(u + i\frac{m}{2}\right)},$$

$$t_m^{\bar{3}}(u) \approx \lim_{\text{TDL}} \frac{Q_2\left(u - i\frac{m}{2}\right)}{Q_2\left(u + i\frac{m}{2}\right)},$$

(115)

which respectively correspond to the $k = 0$, $\ell = 0$ and the $k = m$, $\ell = m$ terms.

Our reasoning is the same as in the previous Sections: we are selecting those terms from the eigenvalues which automatically produce the asymptotic inversions (110). The eigenvalues of the space reflected TM's evaluated at $-u$ are given by formally the same expressions involving the ratios of $Q$-functions, with the signs of the various shifts reversed. After some inspection it can be seen that from the various terms in (113) only the ratios given in (115) lead to the inversion relations. If they would not be the leading terms, then the asymptotic inversion could not hold, and this proves their dominance.

Let us now evaluate the mean values of the charges $X_m(u)$, $Y_m(u)$. We start from a finite volume Bethe state with rapidities $\{\lambda\}_{N_1}, \{\mu\}_{N_2}$.

Assuming that the dominant terms are given by (115) we have for the eigenvalues

$$X_m(u) = -i \lim_{\text{TDL}} \partial_u \log \frac{Q_1\left(u - i\frac{m}{2}\right)}{Q_1\left(u + i\frac{m}{2}\right)} = -i \lim_{\text{TDL}} \sum_{j=1}^{N_1} \left( \frac{1}{u - \lambda_j - i\frac{m}{2}} - \frac{1}{u - \lambda_j + i\frac{m}{2}} \right),$$

$$Y_m(u) = -i \lim_{\text{TDL}} \partial_u \log \frac{Q_2\left(u - i\frac{m}{2}\right)}{Q_2\left(u + i\frac{m}{2}\right)} = -i \lim_{\text{TDL}} \sum_{j=1}^{N_2} \left( \frac{1}{u - \mu_j - i\frac{m}{2}} - \frac{1}{u - \mu_j + i\frac{m}{2}} \right).$$

(116)

These formulas refer to the exact Bethe roots. Making use of the string hypothesis in the TDL we get the expressions

$$X_m(u) = 2\pi L \sum_{n=1}^{\infty} \int_{-\infty}^{\infty} d\lambda \, \rho_n^{(1)}(\lambda) \sum_{j=1}^{\min(m,n)} a_{|n-m|-1+2j}(u - \lambda) =$$

$$= 2\pi \sum_{n=1}^{\infty} \sum_{j=1}^{\min(m,n)} (a_{|n-m|-1+2j} \star \rho_n^{(1)})(u),$$

$$Y_m(u) = 2\pi L \sum_{n=1}^{\infty} \int_{-\infty}^{\infty} d\lambda \, \rho_n^{(2)}(\lambda) \sum_{j=1}^{\min(m,n)} a_{|n-m|-1+2j}(u - \lambda) =$$

$$= 2\pi \sum_{n=1}^{\infty} \sum_{j=1}^{\min(m,n)} (a_{|n-m|-1+2j} \star \rho_n^{(2)})(u).$$

(117)

Here the summation runs over the possible $n$ strings. We made use of the identity

$$\sum_{\ell=1}^{n} \frac{1}{u - \left(\lambda + i\left(\frac{n+1}{2} - \ell\right)\right) - i\frac{m}{2}} - \frac{1}{u - \left(\lambda + i\left(\frac{n+1}{2} - \ell\right)\right) + i\frac{m}{2}} =$$

$$= 2\pi i \sum_{j=1}^{\min(n,m)} a_{|n-m|-1+2j}(u - \lambda).$$

(118)

The formulas (117) can be transformed into more compact forms using standard tricks. This lengthy, but straightforward computation is delegated to appendix B, and the results are the following.

The hole densities can be expressed using the charges as

$$
\begin{aligned}
\rho_{h,m}^{(1)} &= a_m - \frac{1}{2\pi L}\left(X_m^{[+]} + X_m^{[-]} - Y_m\right), \\
\rho_{h,m}^{(2)} &= -\frac{1}{2\pi L}\left(Y_m^{[+]} + Y_m^{[-]} - X_m\right).
\end{aligned}
\tag{119}
$$

Inverting this relation we find

$$
\begin{aligned}
\frac{1}{2\pi L}X_m &= -\left(G_1 \star (\rho_{h,m}^{(1)} + a_m) + G_2 \star \rho_{h,m}^{(2)}\right), \\
\frac{1}{2\pi L}Y_m &= -\left(G_1 \star \rho_{h,m}^{(2)} + G_2 \star (\rho_{h,m}^{(1)} + a_m)\right),
\end{aligned}
\tag{120}
$$

with the kernels

$$
\begin{aligned}
G_1(x) &= \int \frac{dk}{2\pi} e^{-ikx}\hat{G}_1(k) = \frac{1}{\sqrt{3}}\frac{\cosh(\pi/3x)}{\cosh(\pi x)} = \frac{1}{\sqrt{3}}\frac{1}{2\cosh(2\pi x/3)-1}, \\
G_2(x) &= \int \frac{dk}{2\pi} e^{-ikx}\hat{G}_2(k) = \frac{1}{\sqrt{3}}\frac{\sinh(\pi/3x)}{\sinh(\pi x)} = \frac{1}{\sqrt{3}}\frac{1}{2\cosh(2\pi x/3)+1}.
\end{aligned}
\tag{121}
$$

Finally, the root densities can be expressed as

$$
\begin{aligned}
\rho_m^{(1)} &= \frac{1}{2\pi L}\left(X_m^{[+]} + X_m^{[-]} - X_{m-1} - X_{m+1}\right), \\
\rho_m^{(2)} &= \frac{1}{2\pi L}\left(Y_m^{[+]} + Y_m^{[-]} - Y_{m-1} - Y_{m+1}\right).
\end{aligned}
\tag{122}
$$

This is an immediate generalization of the string-charge relation (54) of the $SU(2)$-chain. Once again we can observe the symmetry of the Dynkin diagram of $SU(3)$: exchanging the defining and conjugate representations is mirrored by the exchange of the two Bethe rapidity types.

The crucial point in our derivation was selecting the dominant term in the transfer matrix eigenvalues. The argument based on the asymptotic inversion only holds for the infinite temperature states, and some neighborhood of these root distributions. At present we can not exclude the existence of Bethe root densities, which would select a different term, thus violating (122). Nevertheless we performed an independent check in a particular case, namely for the quench problem with initial state

$$
|\Psi_\delta\rangle = \prod_{j=1}^{L/2} \frac{|11\rangle + |22\rangle + |33\rangle}{\sqrt{3}}.
\tag{123}
$$

This quench was studied in [37, 38] where the exact root densities were determined using Boundary Quantum Transfer Matrix methods. Now we computed the mean values of the first two members $X_1(u)$ and $Y_1(u)$ in this initial state and we checked that the relations (122) indeed hold. This computation is presented in Appendix C.

# 7 Generalization to $SU(N)$

In this section, we consider the generalization of the previous results for $SU(N)$ spin chains. The construction laid out here is a direct generalization of the case of $SU(3)$. However, most of our statements here are conjectures, motivated by the earlier results.

In the $SU(N)$ case the nested Bethe Ansatz involves $(N-1)$ sets of Bethe rapidities, which will be denoted as $\{\{\lambda_j^{(a)}\}_{j=1,\dots,N_a}\}_{a=1,\dots,N-1}$. Correspondingly, the $Q$-functions of the model are

$$Q_0(u) = u^L, \qquad Q_a(u) = \prod_{j=1}^{N_a}(u-\lambda_j^{(a)}), \qquad Q_N(u) = 1. \tag{124}$$

The eigenvalue of the fundamental transfer matrix is

$$t(u) = \sum_{j=1}^{N} z^{(j)}(u), \tag{125}$$

where

$$z^{(\ell)}(u) = \frac{Q_0(u)}{Q_0(u+\mathrm{i})}\frac{Q_{\ell-1}(u+\mathrm{i}\frac{\ell+1}{2})Q_\ell(u+\mathrm{i}\frac{\ell-2}{2})}{Q_{\ell-1}(u+\mathrm{i}\frac{\ell-1}{2})Q_\ell(u+\mathrm{i}\frac{\ell}{2})}, \quad \ell = 1,\dots,N. \tag{126}$$

The Bethe equations are

$$\frac{Q_{\ell-1}(\lambda_j^{(\ell)}+\mathrm{i}\frac{1}{2})}{Q_{\ell-1}(\lambda_j^{(\ell)}-\mathrm{i}\frac{1}{2})}\frac{Q_\ell(\lambda_j^{(\ell)}-\mathrm{i})}{Q_\ell(\lambda_j^{(\ell)}+\mathrm{i})}\frac{Q_{\ell+1}(\lambda_j^{(\ell)}+\mathrm{i}\frac{1}{2})}{Q_{\ell+1}(\lambda_j^{(\ell)}-\mathrm{i}\frac{1}{2})} = -1,$$
$$j = 1\dots N_\ell, \quad \ell = 2,\dots,N-1. \tag{127}$$

With our conventions the 1-string solutions to each nesting level are real rapidities.

Fused transfer matrices $t^\Lambda(u)$ can be constructed for every irreducible representation $\Lambda$ of $SU(N)$ [60, 74], but a special role is played by those representations that correspond to rectangular Young diagrams. For the diagram with $a$ rows and $m$ columns the corresponding transfer matrix is denoted by $t_m^{(a)}(u)$. These objects satisfy the Hirota equation

$$t_m^{(a)}\left(u+\frac{\mathrm{i}}{2}\right)t_m^{(a)}\left(u-\frac{\mathrm{i}}{2}\right) = t_{m+1}^{(a)}(u)t_{m-1}^{(a)}(u) + t_m^{(a-1)}(u)t_m^{(a+1)}(u),$$
$$a = 1,\dots,N-1 \qquad m = 1,2,\dots. \tag{128}$$

We specify the boundary conditions to this system motivated by the $SU(3)$ case: we require

$$t_0^{(a)} = 1, \qquad t_m^{(0)}(u) = \frac{Q_0(u-\mathrm{i}\frac{m}{2})}{Q_0(u+\mathrm{i}\frac{m}{2})}, \qquad t_m^{(N)}(u) = 1. \tag{129}$$

Together with (128) this completely determines the normalization of the local Lax operators and the fused transfer matrices.

For each $t_m^{(a)}(u)$ we define its space reflected variant $\bar{t}_m^{(a)}(u)$. By using local crossing relations and the conjugation properties of $SU(N)$ representations we can express each $\bar{t}_m^{(a)}(u)$ using $t_m^{(N-a)}(u)$; the resulting relations are generalizations of (61) and (97). The space reflected TM's are thus not independent, but the precise relation is irrelevant for our purposes.

Based on the previous results we formulate:

**Conjecture 3.** *For $u \in \mathcal{P}$ the following asymptotic inversion relations hold:*

$$\bar{t}_m^{(a)}(-u)t_m^{(a)}(u) \approx 1. \tag{130}$$

In analogy with the earlier results we introduce the operators

$$X_m^{(a)}(u) = (-\mathrm{i})\partial_u \log t_m^{(a)}(u) \approx (-\mathrm{i})\bar{t}_m^{(a)}(-u)\partial_u t_m^{(a)}(u). \tag{131}$$

**Conjecture 4.** *For $u \in \mathcal{P}$ the traceless operators $\{X_m^{(a)}(u)\}$ are quasi-local.*

The string-charge relations can be established if the eigenvalues of the fused transfer matrices are known. General expressions using Young tableaux were derived in [74], see also Section 7 of the review [63]. The rule to compute the eigenvalues is the following. Let us consider the $(a \times m)$ Young diagram, and all possible semi-standard Young tableaux, i.e. the filling of the diagram with numbers $1, \ldots, N$ such that they are increasing from top to bottom and non-decreasing from left to right. For example, for the Young diagram

$$\begin{array}{|c|c|} \hline & \\ \hline & \\ \hline \end{array} \tag{132}$$

the possible semi-standard tableaux for $N = 3$ are

$$\begin{array}{|c|c|}\hline 1&1\\\hline 2&2\\\hline\end{array},\ \begin{array}{|c|c|}\hline 1&1\\\hline 2&3\\\hline\end{array},\ \begin{array}{|c|c|}\hline 1&1\\\hline 3&3\\\hline\end{array},\ \begin{array}{|c|c|}\hline 1&2\\\hline 2&3\\\hline\end{array},\ \begin{array}{|c|c|}\hline 1&2\\\hline 3&3\\\hline\end{array},\ \begin{array}{|c|c|}\hline 2&2\\\hline 3&3\\\hline\end{array}. \tag{133}$$

In the following let $\tau_{kl}$ denote the element of a tableau $\tau$ in row $k = 1 \ldots a$ and column $l = 1 \ldots m$ from the top left. Then the formula for the eigenvalues is [74]

$$t_m^{(a)}(u) = \prod_{j=1}^{a-1} \frac{Q_0(u + i\frac{m-a+2j}{2})}{Q_0(u - i\frac{m-a+2j}{2})} \times \sum_\tau \left[ \prod_{\substack{k=1\ldots a \\ l=1\ldots m}} z^{(\tau_{kl})}\left(u + i\frac{a-m-2k+2l}{2}\right) \right]. \tag{134}$$

Here the sum runs over all allowed semi-standard tableaux of size $(a \times m)$ for the given $N$, and the $z$-functions are defined in (126). The presence of the pre-factor before the sum is a consequence of our normalization.

In a perhaps more direct way the transfer matrices can be expressed as [74]

$$t_m^{(a)}(u) = \det\left( t_{m-i+j}^{(1)}\left(u + i\frac{i+j-1-a}{2}\right) \right)_{1 \le i,j \le a} \tag{135}$$

$$= \det\left( t_1^{(a-i+j)}\left(u + i\frac{m-i-j+1}{2}\right) \right)_{1 \le i,j \le m}, \tag{136}$$

using the two series $t_m^{(1)}$ or $t_1^{(a)}$, for which we have the formulas

$$t_m^{(1)}(u) = \sum_{1 \le i_1 \le i_2 \le \cdots \le i_m \le N} \left[ \prod_{\ell=1}^m z^{(i_\ell)}\left(u + i\frac{-1-m+2\ell}{2}\right) \right],$$

$$t_1^{(a)}(u) = \frac{Q_0(u + \frac{a-1}{2}i)}{Q_0(u - \frac{a-1}{2}i)} \times \sum_{1 \le i_1 < i_2 < \cdots < i_a \le N} \left[ \prod_{\ell=1}^a z^{(i_\ell)}\left(u + i\frac{a+1-2\ell}{2}\right) \right]. \tag{137}$$

The latter formulas are special cases of the general tableaux sum.

**Theorem 4.** *For each $a = 1 \ldots N-1$ and $m = 1, 2, \ldots$ there is a single term in the expansion* (134) *of $t_m^{(a)}(u)$ which automatically produces the asymptotic inversion* (130). *This term corresponds to the Young tableau where all elements of row $k$ are equal to $k$ for $k = 1, \ldots, a$. The explicit form of this term is*

$$\frac{Q_a\left(u - i\frac{m}{2}\right)}{Q_a\left(u + i\frac{m}{2}\right)}. \tag{138}$$

*Proof.* This can be proven recursively: starting from the bottom right element of the Young diagram, and afterwards considering the elements in the upper rows. If $\tau_{am}$ is the element

in the bottom right corner, then this number can only be present in the bottom row of the diagram. The corresponding $z$-factors will involve the $Q$-functions $Q_0$, $Q_{\tau_{am}-1}$ and $Q_{\tau_{am}}$. The factors of $Q_{\tau_{am}}$ in the product can only come from the boxes filled with $\tau_{am}$, which can occupy a number of cells to the left of the bottom right corner. Collecting these factors we see that the resulting combination of the $Q_{\tau_{am}}$ functions can yield a form satisfying the inversion relation only if $\tau_{am} = a$ and this number occupies the full bottom row; this is a simple consequence of the various shifts present in the $z$-functions and the $Q$-functions. The proof continues by considering the element $\tau_{a-1,m}$ and the corresponding $Q$-functions, showing that the only possibility is that $\tau_{a-1,m} = a - 1$ and this number has to fills the row $a - 1$. This is then repeated for all rows upwards. Collecting all factors of the $Q_0$ functions we see that they just cancel each other, and we obtain (138). □

Using our arguments of the previous Sections it follows that in the thermodynamic limit this term has to be dominant for $u \in \mathcal{P}$:

$$t_m^{(a)}(u) \approx \frac{Q_a\left(u - i\frac{m}{2}\right)}{Q_a\left(u + i\frac{m}{2}\right)}, \qquad a = 1, \dots, N-1, \quad m = 1, 2, \dots \tag{139}$$

A special case of this statement already appeared in a closely related problem in [74]. Note that due to our boundary conditions (129) the above relation holds even for $a = 0$ and $a = N$, using the convention $Q_N(u) = 1$.

Based on (139) and the similarities in the derivation of the TBA equations for all $N$, we propose the following general pattern:

**Conjecture 5.** *In the $SU(N)$-symmetric model the string-charge relations are*

$$\rho_m^{(a)} = \frac{1}{2\pi L}\left(X_m^{(a)[+]} + X_m^{(a)[-]} - X_{m+1}^{(a)} - X_{m-1}^{(a)}\right),$$
$$a = 1 \dots N-1, \qquad m = 1 \dots \infty, \tag{140}$$

*where $\rho_m^{(a)}(u)$ is the density of $m$-strings of rapidity type $a$.*

# 8 Conclusions

We studied the GET (Generalized Eigenstate Thermalization) for higher rank spin models with $SU(N)$, $N \geq 3$ symmetry, with the main focus being on the $N = 3$ Lai-Sutherland model. We argued that a complete set of charges is obtained from the known fusion hierarchy of transfer matrices. These fused transfer matrices correspond to the representations of $SU(3)$ with rectangular diagrams of size $1 \times m$ and $2 \times m$, $m = 1, \dots, \infty$, or equivalently, to symmetrically fused defining and conjugate representations, respectively. We computed the thermodynamic limit of these charges: the resulting string-charge relations take essentially the same form as in the $SU(2)$-invariant XXX chain, with the simple extension of having two particle types and two series of fused charges.

These results in the $SU(3)$ chain possess a conjugation symmetry: exchanging the two families of fused transfer matrices corresponds to exchanging the defining and conjugate representations of $SU(3)$. On the level of the string-charge relations this is reflected by an exchange of the two particle types. The final relations (122) are completely symmetric with respect to this conjugation.

The strictly local charges of the model are only sensitive to the particles of the first type, and their finite volume mean values are computed as

$$\mathcal{Q}_\alpha = \sum_{j=1}^{N_1} q_\alpha(\lambda_j^{(1)}), \tag{141}$$

with $q_\alpha(u)$ being the one-particle eigenvalues. This raises the question: are there local or quasi-local operators which are sensitive only to the second type of particles? Our results show that the $Y_m(u)$ are such operators *in the thermodynamic limit, for almost all Bethe states*. There is no local operator whose finite volume eigenvalues would take the form

$$Y = \sum_{j=1}^{N_2} f(\lambda_j^{(2)}).\tag{142}$$

The string-charge relations are only found in the thermodynamic limit, and hold only for *almost all Bethe states*. The crucial step is the dominance of a prescribed term in the expressions of the transfer matrix eigenvalues.

Based on the derivation for $N = 2, 3$ we conjectured generic results for arbitrary $N$. In particular, we conjectured that the complete GGE is formed by the charges built on the transfer matrices corresponding to the $(a \times m)$ Young diagrams with $a = 1 \ldots N - 1$ and $m = 1 \ldots \infty$.

In some sense these results are not surprising. It is known that in an integrable model with symmetry group $G$ the structure of the nested Bethe Ansatz (for example, the set of the Bethe equations) mirrors the Dynkin diagram of $G$. The fusion rules for the transfer matrices also closely follow the Dynkin diagram. It is thus not surprising that the $T$-system and the set of TBA equations determining the Bethe root densities are so closely related. This correspondence was noted and used in a large number of works already at the end of the 80's and beginning of the 90's; for concrete references see the thorough review [63]. The new addition to the theory was the discovery that in the XXX model the formal expressions of the type $\partial_u \log(t_m(u))$ yield quasi-local operators for $u \in \mathcal{P}$ [25, 26], and that in the TDL they contain just enough information to fix all root densities. What we have performed in this work is to extend this observation to the $SU(N)$-symmetric fundamental models.

In accordance, the crucial points of our work are the proofs of the inversion relations and the quasi-locality. We argued that the quasi-locality property follows once the inversion relation is established on the level of the operators. We computed a detailed proof in two cases, namely for the operators $t_1^{(1)}(u)$ and $t_1^{(2)}(u)$ for $SU(3)$, which correspond to the defining and conjugate representations. Although these are just two particular cases, we believe they constitute strong justification for the remaining conjectures. Also, we remind that even in the $SU(2)$ case complete analytical proofs are available only up to $s = 3/2$ [25, 26].

Naturally, it would be desirable to have explicit proofs in more cases, possibly for the whole fusion hierarchy. On the technical level, the task to be performed is the diagonalization of a 4-site transfer matrix, where these 4 sites carry some fused representations of the symmetry group. Such transfer matrices can be diagonalized by the Bethe Ansatz, and a quite general and completely analytical approach is detailed for example in [75]. It remains to be seen whether this or any alternative techniques prove to be useful for the problem at hand.

Also, it would be interesting to find a more general prescription for the GGE. Based on our results it seems plausible that the string-charge relations are always encoded in the known fusion hierarchy of the theory [61, 63].

# Acknowledgments

The authors would like to thank Tamás Gombor, Enej Ilievski, Márton Mestyán, Junji Suzuki, and Gábor Takács for useful discussions. This research was supported by the BME-Nanotechnology FIKP grant of EMMI (BME FIKP-NAT), by the National Research Development and Innovation Office (NKFIH) (K-2016 grant no. 119204, the OTKA grant no. SNN118028, and the KH-17 grant no. 125567), and by the "Premium" Postdoctoral Program of the Hungarian Academy of Sciences.

# A  Derivation of the asymptotic inversion and the quasi-locality property

We present here the derivation of the asymptotic inversion and quasi-locality property for $t(u)$ and $t^{\bar{3}}(u)$. The two computations are quite similar, using same techniques and the same objects.

We need the adjoints of transfer matrices, hence we start with expressing them.

**Lemma 1.** *The adjoints of the $t, t^{\bar{3}}$ transfer matrices, and their respective space reflected pairs are the following:*

$$t^{\dagger}(u) = \bar{t}(-u^*), \qquad\qquad \left(t^{\bar{3}}\right)^{\dagger}(u) = \bar{t}^{\bar{3}}(-u^*), \tag{143}$$

$$\bar{t}^{\dagger}(u) = t(-u^*), \qquad\qquad \left(\bar{t}^{\bar{3}}\right)^{\dagger}(u) = t^{\bar{3}}(-u^*). \tag{144}$$

*Proof.* Denote by $A^*$ the complex conjugate of $A$, for any matrix or complex number. If $A$ is matrix, the complex conjugation is considered element-wise.

Direct computation shows that the $R$-matrices satisfy:

$$(R(u))^* = R(-u^*), \qquad \left(R^{\bar{3}}(u)\right)^* = R^{\bar{3}}(-u^*). \tag{145}$$

Taking transposition before partial trace reverses the order of $R$-matrices, from where the statement follows. $\qquad\qquad\square$

**Proof of asymptotic inversion.** To prove the (99) asymptotic inversion, we prove the following two statements regarding the norms of the operators:

$$\begin{aligned}
\|t(u)\bar{t}(-u) - 1\|_{\mathrm{HS}}^2 &\approx 0, \\
\|t^{\bar{3}}(u)\bar{t}^{\bar{3}}(-u) - 1\|_{\mathrm{HS}}^2 &\approx 0.
\end{aligned} \tag{146}$$

Expanding the l.h.s. of these expressions leads to the following:

$$\begin{aligned}
\|t(u)\bar{t}(-u) - 1\|_{\mathrm{HS}}^2 &= 3^{-L}\operatorname{Tr} t(u^*)\bar{t}(-u^*)t(u)\bar{t}(-u) - 3^{-L}2\,\Re\operatorname{Tr} t(u)\bar{t}(-u) + 1, \\
\|t^{\bar{3}}(u)\bar{t}^{\bar{3}}(-u) - 1\|_{\mathrm{HS}}^2 &= 3^{-L}\operatorname{Tr} t^{\bar{3}}(u^*)\bar{t}^{\bar{3}}(-u^*)t^{\bar{3}}(u)\bar{t}^{\bar{3}}(-u) - \\
&\qquad\qquad - 3^{-L}2\,\Re\operatorname{Tr} t^{\bar{3}}(u)\bar{t}^{\bar{3}}(-u) + 1.
\end{aligned} \tag{147}$$

As explained in Section 3.3 the traces can be expressed as a partition function of a 2D lattice model, which can be alternatively evaluated by the Quantum Transfer Matrices acting in the "crossed channel". See also [26].

We thus get

$$\begin{aligned}
\|t(u)\bar{t}(-u) - 1\|_{\mathrm{HS}}^2 &= 3^{-L}\operatorname{Tr} t_{AB}(u,u,u^*,u^*)^L - 3^{-L}2\,\Re\operatorname{Tr} t_A(u,u)^L + 1, \\
\|t^{\bar{3}}(u)\bar{t}^{\bar{3}}(-u) - 1\|_{\mathrm{HS}}^2 &= 3^{-L}\operatorname{Tr} \bar{t}_{AB}(u,u,u^*,u^*)^L - 3^{-L}2\,\Re\operatorname{Tr} \bar{t}_A(u,u)^L + 1,
\end{aligned} \tag{148}$$

where

$$\begin{aligned}
t_{AB}(u_1,u_2,v_1,v_2) &= \operatorname{Tr}_a R_{a,1}(v_2)R_{a,2}^t(-v_1)R_{a,1}(u_1)R_{a,2}^t(-u_2), \\
t_A(u_1,u_2) &= \operatorname{Tr}_a R_{a,2}^t(-u_2)R_{a,1}(u_1), \\
t_{AB}^{\bar{3}}(u_1,u_2,v_1,v_2) &= \operatorname{Tr}_a R_{a,1}^{\bar{3}}(v_2)(R_{a,2}^{\bar{3}})^t(-v_1)R_{a,1}^{\bar{3}}(u_1)(R_{a,2}^{\bar{3}})^t(-u_2), \\
t_A^{\bar{3}}(u_1,u_2) &= \operatorname{Tr}_a (R_{a,2}^{\bar{3}})^t(-u_2)R_{a,1}^{\bar{3}}(u_1).
\end{aligned} \tag{149}$$

The traces in (148) can be expressed using the eigenvalues of the above matrices. Let us denote by $\lambda_{AB,j}(u_1, u_2, v_1, v_2)$, $j = 1\ldots 3^4$ and $\lambda_{A,j}(u_1, u_2)$, $j = 1\ldots 3^2$ the eigenvalues of $t_{AB}(u_1, u_2, v_1, v_2)$ and $t_A(u_1, u_2)$, respectively. Similarly, we denote by $\bar{\lambda}_{AB,j}(u_1, u_2, v_1, v_2)$, $j = 1\ldots 3^4$ and $\bar{\lambda}_{A,j}(u_1, u_2)$, $j = 1\ldots 3^2$ the eigenvalues for $\bar{t}_{AB}(u_1, u_2, v_1, v_2)$ and $\bar{t}_A(u_1, u_2)$, respectively.

Then the squared norms are the following:

$$
\begin{aligned}
\|t(u)\bar{t}(-u) - 1\|_{\mathrm{HS}}^2 &= 3^{-L}\sum_{j=1}^{3^4}\lambda_{AB,j}^L(u, u, u^*, u^*) - 3^{-L}2\sum_{j=1}^{3^2}\lambda_{A,j}^L(u, u) + 1, \\
\|t^{\bar{3}}(u)\bar{t}^{\bar{3}}(-u) - 1\|_{\mathrm{HS}}^2 &= 3^{-L}\sum_{j=1}^{3^4}\bar{\lambda}_{AB,j}^L(u, u, u^*, u^*) - 3^{-L}2\sum_{j=1}^{3^2}\lambda_{A,j}^L(u, u) + 1.
\end{aligned}
\tag{150}
$$

As explained in Sec. 3.3, the local inversion relations imply that the delta-states are eigenvectors of the QTM's with trivial eigenvalues equal to 3. If the eigenvalue 3 is non-degenerate and dominant for all of these matrices, then (150) implies (146). Hence, what remains is to prove that 3 is indeed a leading, non-degenerate eigenvalue for all of these matrices, as long as $u \in \mathcal{P}$.

This will be proven somewhat later in this Section. First we consider the quasi-locality property, because its proof also involves the same QTM construction.

**Proof of quasi-locality.** Starting from the definitions

$$
\begin{aligned}
X(u) &= (-\mathrm{i})\bar{t}(-u)\partial\, t(u), \\
Y(u) &= (-\mathrm{i})\bar{t}^{\bar{3}}(-u)\partial\, t^{\bar{3}}(u)
\end{aligned}
\tag{151}
$$

we compute the adjoints of the generator functions:

$$
\begin{aligned}
X^\dagger(u) &= (-\mathrm{i})\left(\partial\,\bar{t}(v)t(-v)\right)\big|_{v=-u^*} = (-\mathrm{i})\left(t(-v)\partial\,\bar{t}(v)\right)\big|_{v=-u^*}, \\
Y^\dagger(u) &= (-\mathrm{i})\left(\partial\,\bar{t}^{\bar{3}}(v)t^{\bar{3}}(-v)\right)\Big|_{v=-u^*} = (-\mathrm{i})\left(t^{\bar{3}}(-v)\partial\,\bar{t}^{\bar{3}}(v)\right)\Big|_{v=-u^*}.
\end{aligned}
\tag{152}
$$

Here we used the commutativity of the transfer matrix with it derivative. Using

$$
\|\{A\}\|_{\mathrm{HS}}^2 = 3^{-L}\mathrm{Tr}\left(A^\dagger A\right) - 3^{-2L}\mathrm{Tr}\left(A^\dagger\right)\mathrm{Tr}(A)
\tag{153}
$$

we get

$$
\begin{aligned}
\|X(u)\|_{\mathrm{HS}}^2 &= -3^{-L}\mathrm{Tr}\, t(u^*)\partial\,\bar{t}(-u^*)\bar{t}(-u)\partial\, t(u) + 3^{-2L}\mathrm{Tr}\, t(u^*)\partial\,\bar{t}(-u^*)\mathrm{Tr}\,\bar{t}(-u)\partial\, t(u), \\
\|Y(u)\|_{\mathrm{HS}}^2 &= -3^{-L}\mathrm{Tr}\, t(u^*)\partial\,\bar{t}(-u^*)\bar{t}(-u)\partial\, t(u) + 3^{-2L}\mathrm{Tr}\, t(u^*)\partial\,\bar{t}(-u^*)\mathrm{Tr}\,\bar{t}(-u)\partial\, t(u).
\end{aligned}
\tag{154}
$$

We consider these traces once more in the rotated channel. In fact, we consider them as two special points of the more general expressions

$$
\begin{aligned}
\mathcal{K}(u_1, u_2, v_1, v_2) &= \partial_{v_1}\partial_{u_1}\Big[3^{-L}\mathrm{Tr}\, t(v_2)\bar{t}(-v_1)\bar{t}(-u_2)t(u_1) - \\
&\qquad\qquad\qquad -3^{-2L}\mathrm{Tr}\, t(v_2)\bar{t}(-v_1)\mathrm{Tr}\,\bar{t}(-u_2)t(u_1)\Big], \\
\bar{\mathcal{K}}(u_1, u_2, v_1, v_2) &= \partial_{v_1}\partial_{u_1}\Big[3^{-L}\mathrm{Tr}\, t^{\bar{3}}(v_2)\bar{t}^{\bar{3}}(-v_1)\bar{t}^{\bar{3}}(-u_2)t^{\bar{3}}(u_1) - \\
&\qquad\qquad\qquad -3^{-2L}\mathrm{Tr}\, t^{\bar{3}}(v_2)\bar{t}^{\bar{3}}(-v_1)\mathrm{Tr}\,\bar{t}^{\bar{3}}(-u_2)t^{\bar{3}}(u_1)\Big],
\end{aligned}
\tag{155}
$$

such that

$$
\|X(u)\|_{\mathrm{HS}}^2 = \mathcal{K}(u, u, u^*, u^*), \qquad\qquad \|Y(u)\|_{\mathrm{HS}}^2 = \bar{\mathcal{K}}(u, u, u^*, u^*).
\tag{156}
$$

We compute $\mathcal{K}$ and $\bar{\mathcal{K}}$ in the rotated channel, similarly as before in the proof of the asymptotic inversion relations. Using the QTM's the quantities $\mathcal{K}$ and $\bar{\mathcal{K}}$ can be expressed as

$$
\begin{aligned}
\mathcal{K}(u_1, u_2, v_1, v_2) &= \partial_{v_1} \partial_{u_1} \big[ 3^{-L} \mathrm{Tr}\, (t_{AB}(u_1, u_2, v_1, v_2))^L - \\
&\qquad\qquad -3^{-2L} \mathrm{Tr}\, (t_A(u_1, u_2))^L \,\mathrm{Tr}\, (t_A(v_1, v_2))^L \big], \\
\bar{\mathcal{K}}(u_1, u_2, v_1, v_2) &= \partial_{v_1} \partial_{u_1} \big[ 3^{-L} \mathrm{Tr}\, \big( t_{AB}^{\bar{3}}(u_1, u_2, v_1, v_2) \big)^L - \\
&\qquad\qquad 3^{-2L} \mathrm{Tr}\, \big( t_A^{\bar{3}}(u_1, u_2) \big)^L \,\mathrm{Tr}\, \big( t_A^{\bar{3}}(v_1, v_2) \big)^L \big].
\end{aligned}
\tag{157}
$$

As before, we consider these expressions in the eigenbases of the respective matrices:

$$
\begin{aligned}
\mathcal{K}(u_1, u_2, v_1, v_2) &= 3^{-L} \sum_{j=1}^{3^4} L \lambda_{AB,j}^{L-1} (\partial_{v_1} \partial_{u_1} \lambda_{AB,j}) + L(L-1) \lambda_{AB,j}^{L-2} (\partial_{v_1} \lambda_{AB,j}) \times \\
&\qquad \times (\partial_{u_1} \lambda_{AB,j}) - 3^{-2L} \sum_{j=1}^{3^2} L \lambda_{A,j}^{L-1} \partial_{u_1} \lambda_{A,j} \sum_{j=1}^{3^2} L \lambda_{A,j}^{L-1} \partial_{v_1} \lambda_{A,j}, \\
\bar{\mathcal{K}}(u_1, u_2, v_1, v_2) &= 3^{-L} \sum_{j=1}^{3^4} L \bar{\lambda}_{AB,j}^{L-1} (\partial_{v_1} \partial_{u_1} \bar{\lambda}_{AB,j}) + L(L-1) \bar{\lambda}_{AB,j}^{L-2} (\partial_{v_1} \bar{\lambda}_{AB,j}) \times \\
&\qquad \times (\partial_{u_1} \bar{\lambda}_{AB,j}) - 3^{-2L} \sum_{j=1}^{3^2} L \bar{\lambda}_{A,j}^{L-1} \partial_{u_1} \bar{\lambda}_{A,j} \sum_{j=1}^{3^2} L \bar{\lambda}_{A,j}^{L-1} \partial_{v_1} \bar{\lambda}_{A,j}.
\end{aligned}
\tag{158}
$$

In the notations above we suppressed the dependence of the eigenvalues on the spectral parameters. For the partial derivatives of $\lambda_{A,j}$, $\bar{\lambda}_{A,j}$, the following arguments are understood:

$$
\partial_{u_1} \lambda_{A,j} \equiv \partial_{u_1} \lambda_{A,j}(u_1, u_2), \qquad\qquad \partial_{v_1} \lambda_{A,j} \equiv \partial_{v_1} \lambda_{A,j}(v_1, v_2), \tag{159}
$$

$$
\partial_{u_1} \bar{\lambda}_{A,j} \equiv \partial_{u_1} \bar{\lambda}_{A,j}(u_1, u_2), \qquad\qquad \partial_{v_1} \bar{\lambda}_{A,j} \equiv \partial_{v_1} \bar{\lambda}_{A,j}(v_1, v_2). \tag{160}
$$

Let us assume that for each matrix above there is a non-degenerate dominant eigenvalue with index $j = 1$. Then the leading terms in the above sums are

$$
\begin{aligned}
\mathcal{K}(u_1, u_2, v_1, v_2) &\approx L^2 \Big( 3^{-L} \lambda_{AB,1}^{L-2} \partial_{u_1} \lambda_{AB,1} \partial_{v_1} \lambda_{AB,1} - 3^{-2L} \lambda_{A,1}^{L-1} \partial_{u_1} \lambda_{A,1} \lambda_{A,1}^{L-1} \partial_{v_1} \lambda_{A,1} \Big) + \\
&+ L \Big( 3^{-L} \lambda_{AB,1}^{L-1} \partial_{v_1} \partial_{u_1} \lambda_{AB,1} - 3^{-L} \lambda_{AB,1}^{L-2} \partial_{v_1} \lambda_{AB,1} \partial_{u_1} \lambda_{AB,1} \Big), \\
\bar{\mathcal{K}}(u_1, u_2, v_1, v_2) &\approx L^2 \Big( 3^{-L} \bar{\lambda}_{AB,1}^{L-2} \partial_{u_1} \bar{\lambda}_{AB,1} \partial_{v_1} \bar{\lambda}_{AB,1} - 3^{-2L} \bar{\lambda}_{A,1}^{L-1} \partial_{u_1} \bar{\lambda}_{A,1} \bar{\lambda}_{A,1}^{L-1} \partial_{v_1} \bar{\lambda}_{A,1} \Big) + \\
&+ L \Big( 3^{-L} \bar{\lambda}_{AB,1}^{L-1} \partial_{v_1} \partial_{u_1} \bar{\lambda}_{AB,1} - 3^{-L} \bar{\lambda}_{AB,1}^{L-2} \partial_{v_1} \bar{\lambda}_{AB,1} \partial_{u_1} \bar{\lambda}_{AB,1} \Big).
\end{aligned}
\tag{161}
$$

The operators $\{X(u)\}$ and $\{Y(u)\}$ are quasi-local if the norm is of $\mathcal{O}(L)$, i.e. the $\mathcal{O}(L^2)$ terms cancel. This will be investigated at the point $u_1 = u_2$, $v_1 = v_2$, which has to be substituted after taking the partial derivatives. As we will see, the eigenvalue 3 is leading and non-degenerate at $u_1 = u_2 = u$, $v_1 = v_2 = u^*$, $u \in \mathcal{P}$, therefore we need to prove that the relations

$$
\begin{aligned}
\partial_{u_1} \bar{\lambda}_{AB,1} \partial_{v_1} \bar{\lambda}_{AB,1} - \partial_{u_1} \bar{\lambda}_{A,1} \partial_{v_1} \bar{\lambda}_{A,1} &= 0, \\
\partial_{u_1} \bar{\lambda}_{AB,1} \partial_{v_1} \bar{\lambda}_{AB,1} - \partial_{u_1} \bar{\lambda}_{A,1} \partial_{v_1} \bar{\lambda}_{A,1} &= 0
\end{aligned}
\tag{162}
$$

hold at $u_1 = u_2$, $v_1 = v_2$. As explained in [26], this follows from the factorizability of the leading eigenvector and the Hellmann-Feynman theorem.

We thus obtain the final result

$$\mathcal{K}(u_1, u_2, v_1, v_2) \approx L\left(\frac{1}{3}\partial_{u_1}\partial_{v_1}\lambda_{AB,1} - \frac{1}{9}\partial_{u_1}\lambda_{AB,1}\partial_{v_1}\lambda_{AB,1}\right),$$
$$\bar{\mathcal{K}}(u_1, u_2, v_1, v_2) \approx L\left(\frac{1}{3}\partial_{u_1}\partial_{v_1}\bar{\lambda}_{AB,1} - \frac{1}{9}\partial_{u_1}\bar{\lambda}_{AB,1}\partial_{v_1}\bar{\lambda}_{AB,1}\right). \tag{163}$$

What remains to be proven is that the delta-states with eigenvalues 3 are indeed dominant for $t_{AB}$, $\bar{t}_{AB}$, $t_A$ and $\bar{t}_A$ and $u \in \mathcal{P}$.

## A.1 The eigensystem of $t_A$ and $\bar{t}_A$

Here we determine the eigenvalues of $t_A$ and $\bar{t}_A$ by a direct computation.

Let us denote the auxiliary spaces of the original physical transfer matrices $t(u)$ and $\bar{t}(u)$ by 0 and $\bar{0}$, respectively. The monodromy matrix in the crossed channel is

$$M_A(u) = R_{A,\bar{0}}^{t_{\bar{0}}}(-u)R_{A,0}(u). \tag{164}$$

Here $A$ stands for the auxiliary space of the rotated transfer matrix, which is identical to the original physical quantum space.

It follows simply from the unitarity relation (13) that the delta-state is an eigenvector of $t_A(u) = \text{Tr}_A M_A(u)$ with eigenvalue 3. Now we compute the full spectrum directly. Writing out the operators and taking the trace in $A$ we get

$$t_A(u) = \frac{-u^2 3 - K_{0\bar{0}}}{-1 - u^2}. \tag{165}$$

The eigenvalues of $K$ are 3 for the trace vector and 0 otherwise, so the eigenvalues are thus

$$3, \qquad 3\frac{u^2}{1 + u^2}. \tag{166}$$

The trace vector is the leading eigenvector whenever $|u^2/(1 + u^2)| < 1$. This is true until $\Re(u^2) > -1/2$. Writing $u = a + ib$ the condition is

$$a^2 - b^2 > -1/2. \tag{167}$$

In the second case, we need to construct the monodromy matrix

$$\bar{M}_A(u) = \bar{R}_{A,0}^{t_0}(-u)\bar{R}_{A,\bar{0}}(u) = \frac{\left(-u + \frac{i}{2}\right)\left(u + \frac{i}{2}\right)}{\left(-u + \frac{3i}{2}\right)\left(u + \frac{3i}{2}\right)} R_{A,0}(u - \sigma)R_{A,\bar{0}}^t(-u - \sigma). \tag{168}$$

More explicitly it reads

$$\bar{M}_A(u) = \frac{\left(u - \frac{i}{2} + P_{A,0}\right)\left(-u - \frac{3i}{2} + iK_{A,\bar{0}}\right)}{\left(-u + \frac{3i}{2}\right)\left(u + \frac{3i}{2}\right)}. \tag{169}$$

The trace becomes

$$t_B(u) = \frac{3\left(\left(\frac{3i}{2}\right)^2 - u^2\right) + 3 - K_{0\bar{0}}}{\left(\frac{3i}{2}\right)^2 - u^2}. \tag{170}$$

Hence the eigenvalues are

$$3, \qquad 3\frac{\frac{3^2}{4} + u^2 - 1}{\frac{3^2}{4} + u^2}. \tag{171}$$

The first one, corresponding to the delta state, is the leading eigenvalue if

$$\Re(u^2) > 1/2 - 3^2/4 = -\frac{7}{4}. \tag{172}$$

Out of the two conditions (167) and (172), the first one is more restricting, and it holds in the physical strip.

## A.2 The eigensystem of $t_{AB}$ and $\bar{t}_{AB}$

Here we compute the spectrum of the 4-site QTM's $t_{AB}(u_1, u_2, v_1, v_2)$ and $\bar{t}_{AB}(u_1, u_2, v_1, v_2)$ at $u_1 = u_2 = u$, $v_1 = v_2 = v$. Both matrices correspond to inhomogeneous, integrable 4-site spin chains, therefore their spectrum can be determined by Bethe Ansatz techniques. One possibility would be the application of the $T-Q$ relations, in the spirit of [75]. However, these are still relatively small matrices of size $81 \times 81$, therefore a direct method is perhaps faster. An exact diagonalization of the matrices is not possible, because the entries are functions of two parameters $u, v$, and the diagonalization lies beyond the capabilities of the symbolic manipulation programs such as Mathematica. Nevertheless, the direct diagonalization is possible after simplifying the matrix using some group theory arguments.

Both transfer matrices are $SU(3)$-invariant, therefore their spectrum can be analyzed by constructing the Clebsch-Gordan series. The transfer matrix $t_{AB}$ acts on the tensor product space

$$3 \otimes \bar{3} \otimes 3 \otimes \bar{3}, \tag{173}$$

whereas $\bar{t}_{AB}$ acts on

$$\bar{3} \otimes 3 \otimes \bar{3} \otimes 3, \tag{174}$$

where we used the standard notations for the defining and conjugate representations. In terms of Young diagrams they are denoted as

$$3 = \square, \qquad \bar{3} = \begin{array}{c}\square\\\square\end{array}. \tag{175}$$

In the following we perform a simple permutation for the vector spaces of $\bar{t}_{AB}$ such that it also acts on the space given by (173).

The Clebsch-Gordan series can be computed with standard methods. We obtain

$$\square \otimes \begin{array}{c}\square\\\square\end{array} \otimes \square \otimes \begin{array}{c}\square\\\square\end{array} = \begin{array}{c}\square\square\square\\\square\end{array} \oplus \square\square\square \oplus \begin{array}{c}\square\square\\\square\square\end{array} \oplus 4 \cdot \begin{array}{c}\square\square\\\square\end{array} \oplus 2 \cdot 1. \tag{176}$$

An easy check-back on the dimensions is

$$3 \cdot 3 \cdot 3 \cdot 3 = 27 + 10 + 10 + 4 \cdot 8 + 2 \cdot 1. \tag{177}$$

Altogether there are 9 irreducible representations, so both $t_{AB}$ and $\bar{t}_{AB}$ can have at most 9 different eigenvalues. Of course, there can be some further degeneracies.

We compute the eigenvalues by focusing on the highest weight states. If there is a representation in the Clebsch-Gordan series with multiplicity one, then the highest weight states have to be eigenstates, and the eigenvalue is found simply by acting with the matrix on the given highest weight state. For the representations with non-trivial multiplicities we have to perform an explicit diagonalization on the finite set of highest weight vectors for that given representation. This will be detailed in the following.

The global $GL(3)$ generators are

$$\Lambda_{jk} = E_{jk}^{(1)} - E_{kj}^{(2)} + E_{jk}^{(3)} - E_{kj}^{(4)}. \tag{178}$$

This follows from the conjugation symmetry. Accordingly, the highest weight vectors in the individual factors in the tensor product (173) can be chosen as

$$|1\rangle_1, \quad |3\rangle_2, \quad |1\rangle_3, \quad |3\rangle_4. \tag{179}$$

Now we consider all components in the Clebsch-Gordan series separately.

-  with dimension 27.

  This representations has multiplicity one, so the highest weight vector has to be an eigenvector of the transfer matrices. The highest weight vector is

$$|1313\rangle. \tag{180}$$

- ☐☐☐ with dimension 10. Once again, this representation is multiplicity free. The highest weight state is

$$|1213\rangle - |1312\rangle. \tag{181}$$

- ☐☐☐☐ with dimension 10, multiplicity free. The highest weight state is:

$$|1323\rangle - |2313\rangle. \tag{182}$$

- 4 copies of ☐☐. Here we have 4 highest weight vectors, and the transfer matrices are closed in the subspace formed by the 4 vectors. A basis in this 4 dimensional space can be formed by taking one delta-state for one product $3 \otimes \bar{3}$ and taking the highest weight vector for the rep ☐☐ in the other product $3 \otimes \bar{3}$. There are indeed 4 ways to do this.

  The basis is thus:

$$
\begin{aligned}
v_1 &= (|11\rangle + |22\rangle + |33\rangle) \otimes |13\rangle, \\
v_2 &= |13\rangle \otimes (|11\rangle + |22\rangle + |33\rangle), \\
v_3 &= |1311\rangle + |2312\rangle + |3313\rangle, \\
v_4 &= |1113\rangle + |1223\rangle + |1333\rangle.
\end{aligned}
\tag{183}
$$

Note that this is not an orthonormal basis. The matrix elements of $t_{AB}$ and $\bar{t}_{AB}$ in this space can be computed by taking into account also the scalar products between the basis vectors. Let $G_{ij} = \langle v_i | v_j \rangle$. Then we have explicitly

$$
G = \begin{pmatrix} 3 & 0 & 1 & 1 \\ 0 & 3 & 1 & 1 \\ 1 & 1 & 3 & 0 \\ 1 & 1 & 0 & 3 \end{pmatrix}. \tag{184}
$$

The actual matrix elements can be computed using the inverse $G^{-1}$, so that within this space we have a 4x4 matrix $\tilde{T}$ such that

$$\tilde{T}_{ij} = G_{ik}^{-1} \langle v_k | T | v_j \rangle. \tag{185}$$

We need to compute the eigenvalues of $\tilde{T}$, which can be done using for example `Mathematica`.

- Finally, there are two singlet representations. A basis is obtained by taking the delta-state $|\delta_{12}\rangle \otimes |\delta_{34}\rangle$ and its permutation, formally written as $|\delta_{14}\rangle \otimes |\delta_{23}\rangle$. The first delta state is an eigenstate, but the permuted one is not. A further simple diagonalization is needed to find the second eigenvector as a linear combination

$$|\delta_{12}\rangle \otimes |\delta_{34}\rangle + \alpha |\delta_{14}\rangle \otimes |\delta_{23}\rangle, \qquad \alpha \in \mathbb{C}. \tag{186}$$

Table 1: Table of the eigenvalues of $t_{AB}$ and $\bar{t}_{AB}$ for the different representations in the Clebsch-Gordan series.

| | $t_{AB}$ | $\bar{t}_{AB}$ |
|---|---|---|
| | $\frac{uv(3uv-2)}{(u^2+1)(v^2+1)}$ | $\frac{48u^2v^2+60(u^2+v^2)-32uv+99}{(4u^2+9)(4v^2+9)}$ |
| | $\frac{3u^2v^2}{(u^2+1)(v^2+1)}$ | $\frac{3\left(16u^2v^2+20(u^2+v^2)+4i(u+v)+9\right)}{(4u^2+9)(4v^2+9)}$ |
| | $\frac{3u^2v^2}{(u^2+1)(v^2+1)}$ | $\frac{3\left(16u^2v^2+20(u^2+v^2)-4i(u+v)+9\right)}{(4u^2+9)(4v^2+9)}$ |
| $4\cdot$ | $\frac{3v^2}{v^2+1}$ | $3-\frac{12}{4v^2+9}$ |
| | $\frac{3u^2}{u^2+1}$ | $3-\frac{12}{4u^2+9}$ |
| | $\frac{3uv(uv+1)}{(u^2+1)(v^2+1)}$ | $\frac{3\left(16u^2v^2+20(u^2+v^2)+16uv-8\sqrt{4-5(u-v)^2}+29\right)}{(4u^2+9)(4v^2+9)}$ |
| | $\frac{3uv(uv+1)}{(u^2+1)(v^2+1)}$ | $\frac{3\left(16u^2v^2+20(u^2+v^2)+16uv+8\sqrt{4-5(u-v)^2}+29\right)}{(4u^2+9)(4v^2+9)}$ |
| $2\cdot 1$ | $3$ | $3$ |
| | $\frac{uv(3uv+2)}{(u^2+1)(v^2+1)}$ | $\frac{3\left(16u^2v^2+20(u^2+v^2)+32uv+65\right)}{(4u^2+9)(4v^2+9)}$ |

The eigenvalues obtained with this method are listed in Table 1. We now analyze the resulting rational functions, and show that the eigenvalue 3 is indeed the dominant one in the physical strip, for both matrices $t_{AB}$ and $\bar{t}_{AB}$.

We are interested in the eigenvalues on the physical strip $\mathcal{P}$ with the restriction $v = u^*$. The main idea to prove that 3 is dominant is to first consider the special point $u = 0$, for which this is seen immediately, and then to determine the algebraic curves on which the magnitudes of the other eigenvalues reach 3. It can be shown that all of these curves are on or outside the boundaries of the physical strip. We will see that in both cases it is the other singlet state which reaches the dominant eigenvalue 3 exactly on the boundary of $\mathcal{P}$.

In the case of $t_{AB}$ six eigenvalues are of the form $\frac{3uv(uv-C)}{(1+u^2)(1+v^2)}$, with $C = 2/3, 0, -1, -2$. The solution to the equation

$$\frac{3uv(uv-C)}{(1+u^2)(1+v^2)} = 3 \tag{187}$$

given that $v = u^*$, $u = a + \mathrm{i}b$ is

$$b = \pm\sqrt{\frac{2+C}{2-C}a^2 + \frac{1}{2-C}}. \tag{188}$$

It can be seen that with these values of $C$ the minimum distance between $u$ and the real axis is always equal to or bigger than $1/2$, thus the eigenvalue crossings do not occur in the physical strip. The other singlet state corresponds to $C = -2$, for which we obtain simply $b = \pm 1/2$: this state becomes degenerate with the delta state exactly at the boundary of the physical strip. The remaining two eigenvalues of $t_{AB}$ can be treated in a similar way, and it can be seen that they do not become dominant within $\mathcal{P}$.

Regarding $\bar{t}_{AB}$ we can see that at $u = v = 0$ 3 is the dominant eigenvalue. Once again we analyze the intersections where the magnitude of the different eigenvalues becomes equal to

3, and we show one by one that these curves lie outside or on the boundary of the physical strip.

- For the first eigenvalue in the Table 1 the substitution $u = v^* = a + ib$ leads to the following equation for the intersection:

$$9 + 8a^2 - 4b^2 = 0. \tag{189}$$

The solution to this equation is $b = \pm\frac{1}{2}\sqrt{8a^2 + 9}$, which shows that the intersection is outside of $\mathcal{P}$.

- The second and third eigenvalues are a complex conjugate pair, hence it is sufficient to consider the absolute value of only one of them. Substituting $u = a + ib$, $v = a - ib$, considering the magnitude of the eigenvalue and setting it equal to 3 leads to a 6th order polynomial equation in $b$. However, it only contains even powers of $b$, hence leading to the following 3rd order equation of the intersection in $B = b^2$:

$$64a^6 + (368 + 64B)a^4 + (620 - 64B^2 - 160B)a^2 + 405 - 64B^3 + 368B^2 - 684B = 0. \tag{190}$$

It can be seen that within the physical strip with $0 \leq B \leq 1/4$ all coefficients of the various powers of $a^2$ are strictly positive, therefore there is no intersection within the physical strip.

- The 4th and 5th eigenvalues also form a complex conjugate pair, hence we only consider the first one. The direct substitution $u = v^* = a + ib$ shows that the magnitudes of these eigenvalues are smaller than 3 for $|b| \leq 1/2$.

- The 6th and 7th eigenvalues on the list are not complex conjugate pairs, but they have a quite similar structure, and we discuss them together. Substitution and simple algebraic manipulation leads to the following equations:

$$4a^2 + 13 = 12b^2 \pm 4\sqrt{5b^2 + 1}. \tag{191}$$

It can be seen that the r.h.s. is always smaller than the minimum of the l.h.s. given by 13, if $|b| \leq 1/2$. Thus there is no intersection in $\mathcal{P}$.

- Finally, regarding the 9th eigenvalue a direct computations shows that this becomes degenerate with 3 if $b = 1/2$, i.e. just on the boundary of the physical strip.

With this we have proven that on the physical strip 3 is indeed a non-degenerate dominant eigenvalue for both $t_{AB}$ and $\bar{t}_{AB}$.

# B TBA derivation of the string-charge relations

Here we compute a compact form for the mean values of the operators $X_m(u)$ and $Y_m(u)$. We start from the expressions (117).

We will use the following convention for the Fourier transform:

$$\hat{f}(k) = \int_{-\infty}^{\infty} du\, f(u)e^{iku}, \tag{192}$$

$$f(u) = \frac{1}{2\pi} \int_{-\infty}^{\infty} dk\, \hat{f}(k)e^{-iku}. \tag{193}$$

For the elementary function $a_n(\lambda)$ defined in (52) we have

$$\hat{a}_n(k) = e^{-\frac{n}{2}|k|}. \tag{194}$$

We need to express

$$\lim_{\text{TDL}} \frac{1}{L} X_m(u) = \sum_{n=1}^{\infty} \int_{-\infty}^{\infty} dk \, \hat{\rho}_n^{(1)}(k) e^{-iku} \sum_{j=1}^{\min(n,m)} \hat{a}_{|n-m|-1+2j}(k), \tag{195}$$

$$\lim_{\text{TDL}} \frac{1}{L} Y_m(u) = \sum_{n=1}^{\infty} \int_{-\infty}^{\infty} dk \, \hat{\rho}_n^{(2)}(k) e^{-iku} \sum_{j=1}^{\min(n,m)} \hat{a}_{|n-m|-1+2j}(k). \tag{196}$$

We apply the summation of the geometric series

$$\sum_{j=1}^{\min(m,n)} \hat{a}_{|n-m|-1+2j}(k) = \sum_{j=1}^{\min(m,n)} e^{-\frac{|k|}{2}(|n-m|-1+2j)} =$$

$$= \frac{1}{2\sinh\left(\frac{|k|}{2}\right)} \left( e^{-\frac{|k|}{2}|n-m|} - e^{-\frac{|k|}{2}(n+m)} \right), \tag{197}$$

to simplify the previous equations to

$$\lim_{\text{TDL}} \frac{1}{L} X_m(u) = \sum_{n=1}^{\infty} \int_{-\infty}^{\infty} dk \, \hat{\rho}_n^{(1)}(k) e^{-iku} \frac{1}{\sinh\left(\frac{|k|}{2}\right)} \left( e^{-\frac{|k|}{2}|n-m|} - e^{-\frac{|k|}{2}(n+m)} \right),$$

$$\lim_{\text{TDL}} \frac{1}{L} Y_m(u) = \sum_{n=1}^{\infty} \int_{-\infty}^{\infty} dk \, \hat{\rho}_n^{(2)}(k) e^{-iku} \frac{1}{\sinh\left(\frac{|k|}{2}\right)} \left( e^{-\frac{|k|}{2}|n-m|} - e^{-\frac{|k|}{2}(n+m)} \right). \tag{198}$$

To proceed, we consider the decoupled TBA equations (84) in Fourier space:

$$\hat{\rho}_{n,t}^{(1)}(k) = \delta_{n,1}\hat{s}(k) + \hat{s}(k)\left( \hat{\rho}_{h,n-1}^{(1)}(k) + \hat{\rho}_{h,n+1}^{(1)}(k) \right) + \hat{s}(k)\hat{\rho}_n^{(2)}(k),$$

$$\hat{\rho}_{n,t}^{(2)}(k) = \hat{s}(k)\left( \hat{\rho}_{h,n-1}^{(2)}(k) + \hat{\rho}_{h,n+1}^{(2)}(k) \right) + \hat{s}(k)\hat{\rho}_n^{(1)}(k), \tag{199}$$

where

$$\hat{s}(k) = \frac{1}{2\cosh\left(\frac{k}{2}\right)}, \qquad \hat{\rho}_{h,0}^{(r)}(k) = 0. \tag{200}$$

For simplicity we will not denote the $k$ argument in the following.

Using the TBA equation in Fourier space, we express the $\rho_m^{(r)}$, $r = 1, 2$ root densities with the $\rho_{h,m}^{(r)}$, $r = 1, 2$ hole densities:

$$\hat{\rho}_n^{(1)} = \frac{1}{1-\hat{s}^2}\left( \hat{s}\left( \delta_{n,1} + \hat{\rho}_{h,n-1}^{(1)} - \frac{1}{\hat{s}}\hat{\rho}_{h,n}^{(1)} + \hat{\rho}_{h,n+1}^{(1)} \right) + \hat{s}^2\left( \hat{\rho}_{h,n-1}^{(2)} - \frac{1}{\hat{s}}\hat{\rho}_{h,n}^{(2)} + \hat{\rho}_{h,n+1}^{(2)} \right) \right),$$

$$\hat{\rho}_n^{(2)} = \frac{1}{1-\hat{s}^2}\left( \hat{s}\left( \hat{\rho}_{h,n-1}^{(2)} - \frac{1}{\hat{s}}\hat{\rho}_{h,n}^{(2)} + \hat{\rho}_{h,n+1}^{(2)} \right) + \hat{s}^2\left( \delta_{n,1} + \hat{\rho}_{h,n-1}^{(1)} - \frac{1}{\hat{s}}\hat{\rho}_{h,n}^{(1)} + \hat{\rho}_{h,n+1}^{(1)} \right) \right). \tag{201}$$

We make use of the following identity:

$$\sum_{n=1}^{\infty} \left( \delta_{n,1} + \hat{\rho}_{n-1,h}^{(r)} - \frac{1}{\hat{s}}\hat{\rho}_{n,h}^{(r)} + \hat{\rho}_{n+1,h}^{(r)} \right) \left( e^{-\frac{|k|}{2}(m+n)} - e^{-\frac{|k|}{2}|m-n|} \right) =$$

$$2\sinh\left(\frac{|k|}{2}\right)\left( \hat{\rho}_{m,h}^{(r)} - e^{-\frac{|k|}{2}m} \right). \tag{202}$$

Substituting this and performing some algebraic manipulations we end up with:

$$
-\frac{1}{2\pi L}\left(2\cosh\left(\frac{k}{2}\right)\hat{X}_m - \hat{Y}_m\right) = \hat{\rho}^{(1)}_{m,h} - e^{-\frac{|k|}{2}m},
$$

$$
-\frac{1}{2\pi L}\left(\cosh\left(\frac{k}{2}\right)\hat{Y}_m - \hat{X}_m\right) = \hat{\rho}^{(2)}_{m,h}.
\tag{203}
$$

After inverse Fourier transformation:

$$
\frac{1}{2\pi L}\left(-X_m\left(u-\frac{i}{2}\right)-X_m\left(u+\frac{i}{2}\right)+Y_m(u)\right) = \rho^{(1)}_{m,h}(u) - a_m(u),
$$

$$
\frac{1}{2\pi L}\left(-Y_m\left(u-\frac{i}{2}\right)-Y_m\left(u+\frac{i}{2}\right)+X_m(u)\right) = \rho^{(2)}_{m,h}(u),
\tag{204}
$$

or in a more compact form:

$$
\rho^{(1)}_{h,m} = a_m - \frac{1}{2\pi L}\left(X^{[+]}_m + X^{[-]}_m - Y_m\right),
$$

$$
\rho^{(1)}_{h,m} = -\frac{1}{2\pi L}\left(Y^{[+]}_m + Y^{[-]}_m - X_m\right).
\tag{205}
$$

This is the first form of our main result, which concerns the hole densities. It is also useful to express the root densities.

Consider (203) and rewrite it in matrix notation:

$$
\begin{pmatrix} 2\cosh(k/2) & -1 \\ -1 & 2\cosh(k/2) \end{pmatrix}\begin{pmatrix} \hat{X}_m \\ \hat{Y}_m \end{pmatrix} = -2\pi L \begin{pmatrix} \hat{\rho}^{(1)}_{h,m} - e^{-\frac{|k|}{2}m} \\ \hat{\rho}^{(2)}_{h,m} \end{pmatrix}.
\tag{206}
$$

Consider the inverse matrix,

$$
\frac{1}{e^k + e^{-k} + 1}\begin{pmatrix} 2\cosh(k/2) & 1 \\ 1 & 2\cosh(k/2) \end{pmatrix}.
\tag{207}
$$

With the help of it we can express the charges with the root densities:

$$
X_m = -2\pi L\left(G_1 \star (\rho^{(1)}_{h,m} + a_m) + G_2 \star \rho^{(2)}_{h,m}\right),
$$

$$
Y_m = -2\pi L\left(G_1 \star \rho^{(2)}_{h,m} + G_2 \star (\rho^{(1)}_{h,m} + a_m)\right),
\tag{208}
$$

where $G_1(u)$, $G_2(u)$ are the inverse transformed versions of

$$
\hat{G}_1(k) = \frac{e^{k/2} + e^{-k/2}}{e^k + e^{-k} + 1},
$$

$$
\hat{G}_2(k) = \frac{1}{e^k + e^{-k} + 1},
\tag{209}
$$

and their explicit form can be computed using standard techniques:

$$
G_1(x) = \int \frac{dk}{2\pi} e^{-ikx}\hat{G}_1(k) = \frac{1}{\sqrt{3}}\frac{\cosh(\pi/3x)}{\cosh(\pi x)} = \frac{1}{\sqrt{3}}\frac{1}{2\cosh(2\pi x/3)-1},
$$

$$
G_2(x) = \int \frac{dk}{2\pi} e^{-ikx}\hat{G}_2(k) = \frac{1}{\sqrt{3}}\frac{\sinh(\pi/3x)}{\sinh(\pi x)} = \frac{1}{\sqrt{3}}\frac{1}{2\cosh(2\pi x/3)+1}.
\tag{210}
$$

Consider the (84) TBA equation, and after Fourier transformation, construct a similar matrix form out of it:

$$\begin{pmatrix} 1 & -\hat{s} \\ -\hat{s} & 1 \end{pmatrix} \begin{pmatrix} \hat{\rho}_{t,m}^{(1)} \\ \hat{\rho}_{t,m}^{(2)} \end{pmatrix} = \begin{pmatrix} \delta_{m,1}\hat{s} + \hat{s}\left(\rho_{h,m-1}^{(1)} + \rho_{h,m+1}^{(1)} - \rho_{h,m}^{(2)}\right) \\ \hat{s}\left(\rho_{h,m-1}^{(2)} + \rho_{h,m+1}^{(2)} - \rho_{h,m}^{(1)}\right) \end{pmatrix}. \tag{211}$$

After taking the inverse matrix Fourier inverse transformation, we arrive at the following form:

$$\rho_{t,m}^{(1)} = \delta_{m,1}G_1 + G_1 \star (\rho_{h,m-1}^{(1)} + \rho_{h,m+1}^{(1)} - \rho_{h,m}^{(2)}) + G_2 \star (\rho_{h,m-1}^{(2)} + \rho_{h,m+1}^{(2)} - \rho_{h,m}^{(1)}),$$
$$\rho_{t,m}^{(2)} = \delta_{m,1}G_2 + G_1 \star (\rho_{h,m-1}^{(2)} + \rho_{h,m+1}^{(2)} - \rho_{h,m}^{(1)}) + G_2 \star (\rho_{h,m-1}^{(1)} + \rho_{h,m+1}^{(1)} - \rho_{h,m}^{(2)}). \tag{212}$$

Using (208) and (204) we get

$$\rho_m^{(1)} = \delta_{m,1}G_1 - \frac{1}{2\pi L}\left(X_{m-1} + X_{m+1} - X_m^{[+]} - X_m^{[-]}\right) - G_1 \star (a_{m-1} + a_{m+1}) + G_2 \star a_m - a_m,$$
$$\rho_m^{(2)} = \delta_{m,1}G_2 - \frac{1}{2\pi L}\left(Y_{m-1} + Y_{m+1} - Y_m^{[+]} - Y_m^{[-]}\right) - G_2 \star (a_{m-1} + a_{m+1}) + G_1 \star a_m. \tag{213}$$

Making use of the identities:

$$s \star (a_{n-1} + a_{n+1}) = a_n, \qquad n > 1$$
$$s \star a_2 = a_1 + s,$$
$$G_2 = G_1 \star s,$$
$$-G_1 \star a_2 + G_2 \star a_1 + a_1 = -G_1, \tag{214}$$

we express the string-charge relations in a uniform, source term free way:

$$\rho_m^{(1)} = \frac{1}{2\pi L}\left(X_m^{[+]} + X_m^{[-]} - X_{m-1} - X_{m+1}\right),$$
$$\rho_m^{(2)} = \frac{1}{2\pi L}\left(Y_m^{[+]} + Y_m^{[-]} - Y_{m-1} - Y_{m+1}\right). \tag{215}$$

## C  Checking the string-charge relations for a particular quench

Let us consider the quantum quench in the $SU(3)$-invariant model with the specific initial state

$$|\Psi_\delta\rangle = \prod_{j=1}^{L/2} \frac{|11\rangle + |22\rangle + |33\rangle}{\sqrt{3}}. \tag{216}$$

This quench has been treated in detail in the works [37,38] using Boundary Quantum Transfer Matrix techniques. In particular, the following exact results were computed there.

In the long time limit the system is populated by Bethe states with root densities $\{\rho_m^{(1)}(u), \rho_m^{(2)}(u)\}$, such that the total densities (sums of the densities of roots and holes) $\rho_{t,m}^{(a)}(u) = \rho_m^{(a)}(u) + \rho_{h,m}^{(a)}(u)$ are

$$\rho_{t,1}^{(1)}(u) = \frac{1}{2\pi} \frac{16(80u^4 + 168u^2 + 53)}{(4u^2 + 1)(8u^2 + 3)(4u^2 + 9)^2},$$
$$\rho_{t,1}^{(2)}(u) = \frac{1}{2\pi} \frac{(4u^2 + 1)(5u^4 + 18u^2 + 8)}{(u^2 + 1)^2(u^2 + 4)^2(8u^2 + 3)}. \tag{217}$$

The ratios of root and hole densities are given by the so-called $Y$ functions defined as

$$\eta_m^{(a)}(u) = \frac{\rho_{h,m}^{(a)}(u)}{\rho_m^{(a)}(u)}. \tag{218}$$

We use this notation in order to avoid confusion with the $Y(u)$ charge operators.

The exact $Y$-functions for the 1-strings of the first and second type were computed as

$$1 + \eta_1^{(1)}(u) = 1 + \eta_1^{(2)}(u) = \frac{3(4u^2 + 1)}{4u^2}. \tag{219}$$

From the above equations we can compute the first two hole densities as

$$\rho_{h,1}^{(1)} = \frac{\eta_1^{(1)}(u)}{1 + \eta_1^{(1)}(u)} \rho_1^{(1)} = \frac{1}{2\pi} \frac{16(80u^4 + 168u^2 + 53)}{3(4u^2 + 1)^2(4u^2 + 9)^2},$$

$$\rho_{h,1}^{(2)} = \frac{\eta_1^{(2)}(u)}{1 + \eta_1^{(2)}(u)} \rho_1^{(2)} = \frac{1}{2\pi} \frac{(5u^4 + 18u^2 + 8)}{3(u^2 + 1)^2(u^2 + 4)^2}. \tag{220}$$

In the following we compute the mean values of the charges $X_1(u)$ and $Y_1(u)$ in $|\Psi_\delta\rangle$ using their definition:

$$X_1(u) = -i \left. \frac{\partial}{\partial \lambda} \langle \Psi_\delta | \bar{t}(-u) t(\lambda) | \Psi_\delta \rangle \right|_{u=\lambda},$$

$$Y_1(u) = -i \left. \frac{\partial}{\partial \lambda} \langle \Psi_\delta | (\bar{t}^{\bar{3}}(-u) t^{\bar{3}}(\lambda) | \Psi_\delta \rangle \right|_{u=\lambda}. \tag{221}$$

From this we will compute the hole densities directly from the string-charge relations (119), which will be compared to (220).

The above mean values can be evaluated using standard methods, by building the corresponding 2D partition functions, and evaluating them with double row transfer matrices in the crossed channel [21, 37, 38], see also Fig. 5. In the TDL we have

$$\langle \Psi_\delta | \bar{t}(-u) t(\lambda) | \Psi_\delta \rangle \to \left( \Lambda_\delta^{(1)}(\lambda, u) \right)^{L/2},$$

$$\langle \Psi_\delta | (\bar{t}^{\bar{3}}(-u) t^{\bar{3}}(\lambda) | \Psi_\delta \rangle \to \left( \Lambda_\delta^{(2)}(\lambda, u) \right)^{L/2}, \tag{222}$$

where $\Lambda_\delta^{(1,2)}(\lambda, u)$ are the leading eigenvalues of the corresponding double row QTM's.

The double row QTM's can be diagonalized in a relatively simple way. Due to the boundary conditions they are only $SO(3)$-symmetric, and the 9 dimensional Hilbert space on which they act splits into the $SO(3)$-representations

$$3 \otimes 3 = 5 + 3 + 1. \tag{223}$$

The singlet representation corresponds to the delta state, and it follows from the inversion relations that at $u = \lambda$ the corresponding eigenvalue is

$$\Lambda_\delta^{(1)}(\lambda, \lambda) = \Lambda_\delta^{(2)}(\lambda, \lambda) = 1. \tag{224}$$

We checked that the other two eigenvalues are indeed sub-leading in the physical strip, for both QTM's.

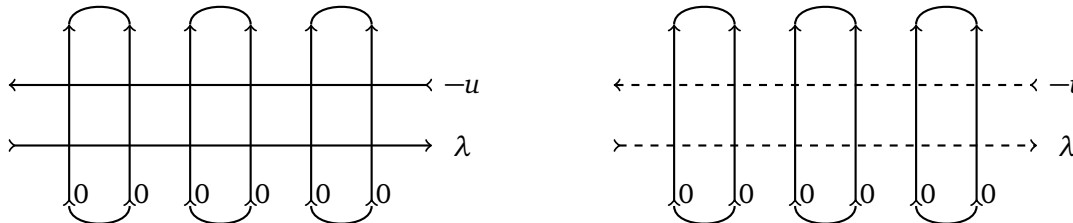

Figure 5: Evaluation of the mean values of $X_1(u)$ and $Y_1(u)$ in the delta state. The solid and dashed lines stands for auxiliary spaces carrying the defining and conjugate representations of $SU(3)$, respectively, and the arrows denote the direction of the action of the $R$ matrices. The horizontal lines with spectral parameters $\lambda$ and $u$ stem from the action of the various transfer matrices. The vertical lines correspond to the physical spaces of the homogeneous chain, thus their spectral parameter is equal to zero. We have periodic boundary conditions in the horizontal direction. These partition functions can be evaluated in the crossed channel, by building the QTM's which act from the left to the right.

$A(\lambda, u):$                                        $B(\lambda, u):$

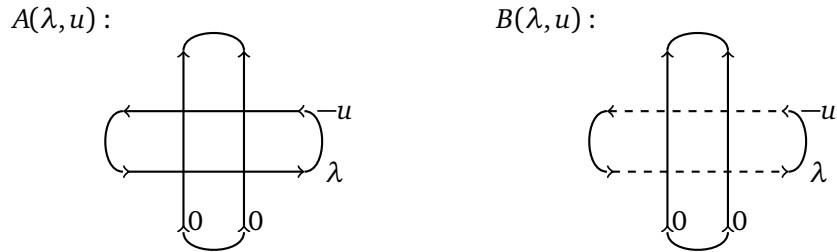

Figure 6: Evaluation of the leading eigenvalues of the QTM's, which belong to the delta state in the crossed channel. The eigenvalue is computed by sandwiching the QTM between the eigenvector from the left and right. We thus obtain the partition functions above, where the boundary conditions are given by the delta states in all 4 directions. These partition functions can then be evaluated as a single trace, which is obtained for example in an anti-clockwise manner leading to the expressions (226).

The mean values of the $X_1(u)$ and $Y_1(u)$ operators are thus

$$X_1(u) = -\frac{iL}{2}\frac{\partial}{\partial\lambda}\Lambda_\delta^{(1)}(\lambda, u)\Big|_{u=\lambda},$$
$$Y_1(u) = -\frac{iL}{2}\frac{\partial}{\partial\lambda}\Lambda_\delta^{(2)}(\lambda, u)\Big|_{u=\lambda}. \tag{225}$$

Even though the simple result (224) holds at $u = \lambda$, the leading eigenvalue is some non-trivial rational function at $u \neq \lambda$, which we now compute. The simplest way is perhaps to draw the partition function corresponding to the eigenvalue, and to evaluate it as a single trace, see Fig. 6. This leads to

$$X_1(u) = -\frac{iL}{2}\frac{1}{9}\partial_\lambda\mathrm{Tr}\left(R^t(\lambda)R^t(-u)R(-u)R(\lambda)\right)\Big|_{u=\lambda},$$
$$Y_1(u) = -\frac{iL}{2}\frac{1}{9}\partial_\lambda\mathrm{Tr}\left(\bar{R}^t(\lambda)\bar{R}^t(-u)\bar{R}(-u)\bar{R}(\lambda)\right)\Big|_{u=\lambda}. \tag{226}$$

Here the extra normalization factor of 1/9 comes from the normalization of the physical delta states.

Using the definitions (11)-(90) we have to compute the quantities

$$
\begin{aligned}
A(\lambda, u) &\equiv \mathrm{Tr}\left(R^t(\lambda)R^t(-u)R(-u)R(\lambda)\right) = \\
&= \frac{\mathrm{Tr}\left((\lambda + iK)(-u + iK)(-u + iP)(\lambda + iP)\right)}{(\lambda + i)^2(-u + i)^2}, \\
B(\lambda, u) &\equiv \mathrm{Tr}\left(\bar{R}^t(\lambda)\bar{R}^t(-u)\bar{R}(-u)\bar{R}(\lambda)\right) = \\
&= \frac{\mathrm{Tr}\left((\lambda + i\frac{3}{2} - iP)(-u + i\frac{3}{2} - iP)(-u + i\frac{3}{2} - iK)(\lambda + i\frac{3}{2} - iK)\right)}{(\lambda + i\frac{3}{2})^2(-u + i\frac{3}{2})^2}.
\end{aligned}
\tag{227}
$$

Direct computation of the traces gives

$$
A(\lambda, u) = 3\frac{3 - (4i)\lambda - \lambda^2 + (4i)u + 8\lambda u - (2i)\lambda^2 u - u^2 + (2i)\lambda u^2 + 3\lambda^2 u^2}{(\lambda + i)^2(-u + i)^2}
\tag{228}
$$

and

$$
B(\lambda, u) = \frac{3}{16}\frac{75 - (188i)\lambda - 172\lambda^2 - (172i)u - 304\lambda u + (176i)\lambda^2 u - 76u^2 + (112i)\lambda u^2 + 48\lambda^2 u^2}{(\lambda + i\frac{3}{2})^2(-u + i\frac{3}{2})^2}.
\tag{229}
$$

For the derivatives we get

$$
\begin{aligned}
X_1(\lambda) &= \frac{-iL}{18}\partial_\lambda A(\lambda, u)\Big|_{u=\lambda} = \frac{1}{3}\frac{1 + 2\lambda^2}{(\lambda^2 + 1)^2}, \\
Y_1(\lambda) &= \frac{-iL}{18}\partial_\lambda B(\lambda, u)\Big|_{u=\lambda} = \frac{4}{3}\frac{5 + 4\lambda^2}{(4\lambda^2 + 9)^2}.
\end{aligned}
\tag{230}
$$

And finally, we compute the one-string hole densities via

$$
\begin{aligned}
\rho_{h,1}^{(1)} &= \frac{1}{2\pi L}\left[X_1^{[+]} + X_1^{[-]} - Y_1\right] - a_1, \\
\rho_{h,1}^{(2)} &= \frac{1}{2\pi L}\left[Y_1^{[+]} + Y_1^{[-]} - X_1\right],
\end{aligned}
\tag{231}
$$

where

$$
a_1 = \frac{1}{2\pi}\frac{1}{u^2 + 1/4}.
\tag{232}
$$

After substitution we get the same results (220) as obtained previously.

It is important that our check is independent from the derivation of [37, 38], which was built on the fusion hierarchy of the Boundary QTM's. Even though the methods of [37, 38] also involved double row, two-site transfer matrices, a close inspection shows that the actual construction there is different, for example the rapidities involved are chosen in a different way. Our present check is thus an independent confirmation of the string-charge relations in the $SU(3)$-symmetric chain.

# D   Proof of the local inversion relations

Here we perform an explicit computation of the product

$$
R^\Lambda(u)R^\Lambda(-u),
\tag{233}
$$

where $\Lambda$ is an irreducible representation of $GL(N)$ described by a rectangular Young diagram, and the $R$-matrix is given generally as

$$
R^\Lambda(u) = \frac{u + i\alpha + iE_{ij}\Lambda_{ji}}{u + i\alpha'},
\tag{234}
$$

with some shift parameters $\alpha, \alpha' \in \mathbb{R}$.

The $R$-matrix acts on the tensor product of two representations. Let us consider the Clebsch-Gordon series

$$\Lambda_1 \otimes \Lambda = \oplus_k \Lambda_k, \tag{235}$$

where $\Lambda_1$ is the defining representation. Generally the $R$-matrix can be decomposed as

$$R^\Lambda(u) = \sum_k \rho_k(u) P_k, \tag{236}$$

where $P_k$ are the projectors onto the invariant subspaces. It follows that the inversion relation can be satisfied if

$$\rho_k(u)\rho_k(-u) = 1 \tag{237}$$

holds for all components.

In order to compute $\rho_k(u)$ we first compute the eigenvalues of $E_{ij}\Lambda_{ji}$ which can be expressed using quadratic Casimir operators

$$C_2 = \Lambda_{ij}\Lambda_{ji} \tag{238}$$

as

$$E_{ij}\Lambda_{ji} = \frac{1}{2}\left((E_{ij} + \Lambda_{ij})(E_{ji} + \Lambda_{ji}) - E_{ij}E_{ji} - \Lambda_{ij}\Lambda_{ji}\right). \tag{239}$$

The quadratic Casimir for the representation $\Lambda$ of $GL(N)$ with highest weight $(h_1, \ldots, h_N)$ is

$$C_2 = \sum_{j=1}^N h_j^2 + \sum_{j<k}(h_j - h_k). \tag{240}$$

We consider some examples of this formula. For the defining representation the highest weight is $(1, 0, \ldots, 0)$ and the Casimir is

$$C_2 = N. \tag{241}$$

For the symmetrically fused representation with highest weight $(m, 0, 0, \ldots)$ (corresponding to the $(1 \times m)$ Young diagram)

$$C_2 = m(m + N - 1). \tag{242}$$

For the anti-symmetric tensor with highest weight $(1, 1, 0, \ldots)$ (corresponding to the $(2 \times 1)$ Young diagram)

$$C_2 = 2N - 3. \tag{243}$$

For symmetrically fused anti-symmetric representations with highest weight $(m, m, 0, \ldots)$ (corresponding to the $(2 \times m)$ Young diagram)

$$C_2 = 2m(m + N - 2). \tag{244}$$

Let us now focus on the $GL(3)$ representations described by rectangular Young diagrams.

- Let us take $(m, 0, 0)$. The Clebsch Gordan series (235) has two terms:

$$(1, 0, 0) \otimes (m, 0, 0) = (m + 1, 0, 0) \oplus (m, 1, 0). \tag{245}$$

  For the component $(m + 1, 0, 0)$ we have

$$E_{ij}\Lambda_{ji} = m. \tag{246}$$

  For the component $(m, 1, 0)$ we have

$$E_{ij}\Lambda_{ji} = -1. \tag{247}$$

It follows that the eigenvalues of the $R$-matrix

$$R^{(m,0)}(u) = \frac{u - \mathrm{i}\frac{m-1}{2} + \mathrm{i}E_{ij}\Lambda_{ji}}{u + \mathrm{i}\frac{m+1}{2}} \tag{248}$$

are

$$\frac{u - \mathrm{i}\frac{m-1}{2} + \mathrm{i}m}{u + \mathrm{i}\frac{m+1}{2}} = 1, \qquad \frac{u - \mathrm{i}\frac{m-1}{2} - \mathrm{i}}{u + \mathrm{i}\frac{m+1}{2}} = \frac{u - \mathrm{i}\frac{m+1}{2}}{u + \mathrm{i}\frac{m+1}{2}}. \tag{249}$$

- Let us now take $(m, m, 0)$. There are two components in the Clebsch-Gordan series (235):

$$(1, 0, 0) \otimes (m, m, 0) = (m + 1, m, 0) \oplus (m, m, 1). \tag{250}$$

For the component $(m + 1, m, 0)$ we have

$$E_{ij}\Lambda_{ji} = m, \tag{251}$$

whereas for $(m, m, 1, 0)$ we have

$$E_{ij}\Lambda_{ji} = -2. \tag{252}$$

It follows that the eigenvalues of

$$R^{(1,0),(0,m)}(u) = \frac{u - \mathrm{i}\frac{m-2}{2} + \mathrm{i}E_{ij}\Lambda_{ji}}{u + \mathrm{i}\frac{m+2}{2}} \tag{253}$$

are

$$\frac{u - \mathrm{i}\frac{m-2}{2} + \mathrm{i}m}{u + \mathrm{i}\frac{m+2}{2}} = 1, \qquad \frac{u - \mathrm{i}\frac{m-2}{2} - 2\mathrm{i}}{u + \mathrm{i}\frac{m+2}{2}} = \frac{u - \mathrm{i}\frac{m+2}{2}}{u + \mathrm{i}\frac{m+2}{2}}. \tag{254}$$

We can see that both $R$-matrices satisfy the local inversion relation.

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
