# Peer review of "Generalized Gibbs Ensemble and string-charge relations in nested Bethe Ansatz"

_SciPost Physics, doi:SciPost Phys. 8, 034 (2020)_

## Round 2 · Referee Report · Anonymous (Referee 1) · 2019-9-26

Strengths

The manuscript is very thorough, and the derivations are long and clear. It is written in a way suggesting further generalizations and future applications to other Bethe-Ansatz-solvable models.

Weaknesses

Nothing significant.

Report

The paper analyses the recently discovered phenomenon of an incompleteness, in the thermodynamic limit, of the set of local conserved quantities as a set of operators entering the sum the logarithm of the Generalized Gibbs density matrix. The models authors use as a testing ground---the generalized Heisenberg chains---are, probably, the most ideally suited to illustrate the phenomenon and suggest remedies. For these models, authors obtain explicit expressions for the additional, now only quasi-local charges that complete the Generalized Gibbs exponent. Thanks to the Bethe Ansatz integrability of the models considered, the criterion for completeness is quite transparent: the expressibility of the rapidity density through the charges of the alleged complete set alone.

The paper is well written, it deserves publishing inits current form.

Requested changes

Optionally, I would at least consider referencing [V. E. Korepin, N. M. Bogoliubov, A. G. Izergin,
"Quantum Inverse Scattering Method and Correlation Functions"]: the thing authors call a Generalized Eigenstate Thermalization" is present there under a "representative state hypothesis".

---

## Round 2 · Referee Report · Anonymous (Referee 2) · 2019-10-15

Strengths

technically strong and precise
well written

Weaknesses

using previously introduced techniques and concepts
some of the more general statements do not seem to be accurate (see report)

Report

In this paper, the authors propose the exact correspondence between quasi-local charges and thermodynamic Bethe ansatz in the SU(N) integrable model. This involves the nested Bethe ansatz, and in that respect goes further than previous works on the subject. The paper proposes that a certain set of conserved charges, built out of transfer matrices based on higher-spin auxiliary-space representations, are quasi-local, and that their eigenvalues fix the full string distributions, which arise from the string hypothesis, of the nested Bethe ansatz. Certain aspects are proven rigorously, such as the quasi-locality of some of the charges.

The paper is professionally and clearly written, and the results are strong and, as far as I can see, correct. I think this is an excellent work, and I accept it for publication.

I have only two more philosophical comments in relation to statements made by the authors in the introduction, and I hope the authors can clarify their comments accordingly.

First, on page 4, it is mentioned that the complete GGE as written in eq 1.8 can never be quasi-local. This is not true. I take here that the authors use the definition of quasi-local in 2.1; this was, previously, referred to as “pseudolocal” (quasi-local being reserved for densities with exponentially decaying envelope). It was shown in [B. Doyon, Thermalization and pseudolocality in extended quantum systems, Commun. Math. Phys. 351, 155 (2017)], mathematically rigorously, that if the large-time limit exists (that is, there is relaxation, in a given precise sense), then, from a large family of states, including thermal states of arbitrary local hamiltonians, the state that comes out of the quench is pseudolocal (quasi-local in the author’s definition). That is, it is essentially of the form 1.8, with the result of the series $\sum_i \beta_i Q_i$ being a pseudolocal charge. For instance, the use of the mode operators in free theories, $\int dp f(p) a^\dagger(p)a(p)$, does not preclude pseudolocality, all details are in the properties of the coefficient $f(p)$. Not all possible functions $f(p)$ can occur - pseudolocality imposes certain conditions. This is a theorem - it comes with conditions of relaxation, and it has a precise expression which is a bit more involved than expression 1.8. If the authors know for sure that pseudolocality is broken, they should mention how the conditions of the theorem are actually broken and explain a bit more.

Second, on page 4 again, it is mentioned that the complete GGE does not use the maximum entropy principle. Again, I disagree, at least with the philosophy of this statement. I agree that in order to determine the $\beta_i$ in 1.8 - or, equivalently, the root densities - in any given quench, one does not use explicitly a maximisation of entropy; this principle is used in a different method, the quench action method. However, on principles, both method are equivalent: in the method fixing the beta’s (or root densities) from evaluating averages of conserved charges in the initial state, one uses maximal-entropy *implicitly*. The form 1.8, or its equivalent in terms of distribution of Bethe roots, is a state obtained by *maximising entropy with respect to the conditions on the conserved densities*. It is, like the Gibbs state, a state obtained by maximising entropy. The use of this form in quench protocols already implies that we assume that entropy is maximised after long times. And indeed, a large amount of information is lost in this process - one cannot reconstruct the exact initial state. That the GGE fixes all Bethe root densities does not preclude the fact that entropy has been maximised.

Finally, on pages 6,7, of course there is a more general notion of pseudolocality (quasi-locality here), with respect to other states, see []; this in fact also something alluded to by the author later on in the paper.

Requested changes

adjust statements as per the three points in the report.

  • validity: top
  • significance: high
  • originality: ok
  • clarity: top
  • formatting: perfect
  • grammar: perfect

Author:  Balázs Pozsgay  on 2020-02-10  [id 734]

(in reply to Report 2 on 2019-10-15)

We are thankful to the Referees for their comments. We would like to reply to this Report of Referee 1. in detail.

  1. The referee pointed out that in [Commun. Math. Phys. 351, 155 (2017)] it was proven that after equilibration the emerging ensembles are actually quasi-local, if some reasonable requirements are fulfilled. We admit we did not know this statement, therefore our comments in the manuscript were superficial. A thorough analysis of this issue is beyond the scope of this work, so we decided to delete the corresponding sentences from the manuscript. This way we do not treat this question, which was not a main question for this work anyway. We uploaded a new version of the manuscript to arxiv, and we are willing to resubmit here (there has not yet been an editorial decision to do so).

  2. Regarding the second question, we agree that this is more ``philosophical''. Also, we do not completely agree with the referee, or perhaps there is still some misunderstanding. If the mean charges are completely specified, then the string-charge relations completely determine the string densities. The entropy principle is used implicitly only in the sense that we are looking for state with well defined string densities, which is an overwhelming majority, and we are not looking at any kind of improbable outlier states. So for this question we decided to keep our comments in the way they were given first. We hope the referee can agree to this.

---

## Round 2 · Author Response

We noticed that we made a mistake regarding the inversion of $R$-matrices. The mistake is irrelevant for our conclusions, because it concerns representations that are NOT used in the paper. Nevertheless we corrected the mistake.

---

## Round 2 · List of Changes

We had two statements regarding the local inversion relations. The first said that for representations with rectangular Young diagrams the $R$-matrices are linear and satisfy the inversion. This is correct. The second statement was that for other representations the $R$-matrices are still linear, but they do not satisfy the inversion. This is not true: for other representations the $R$-matrices still satisfy the local inversion, but they are not linear. We now corrected this, and added the references from which this becomes clear.

---

## Editorial Decision

published